

# Comparison of dust layer heights from active and passive satellite sensors

Arve Kylling[1], Sophie Vandenbussche[2], Virginie Capelle[3], Juan Cuesta[4], Lars Klüser[5], Luca Lelli[6], Thomas Popp[5], Kerstin Stebel[1], and Pepijn Veefkind[7,8]

[1]NILU - Norwegian Institute for Air Research, PO Box 100, 2027 Kjeller, Norway
[2]Royal Belgian Institute for Space Aeronomy (BIRA-IASB), Brussels, Belgium
[3]Laboratoire de Météorologie Dynamique (LMD), UMR8539, CNRS/IPSL, Ecole Polytechnique, Palaiseau, France
[4]Laboratoire Interuniversitaire des Systémes Atmosphériques (LISA), CNRS UMR7583, Université Paris Est Créteil, Université Paris Diderot, Créteil, France
[5]Deutsches Zentrum für Luft-und Raumfahrt e. V. (DLR), Deutsches Fernerkundungsdatenzentrum (DFD), 82234 Oberpfaffenhofen, Germany
[6]Institute of Environmental Physics (IUP), University of Bremen, Bremen, Germany
[7]Royal Netherlands Meteorological Institute (KNMI), 3730 AE De Bilt, The Netherlands
[8]Geosciences and Remote Sensing, Delft University of Technology, 2628 AA Delft, The Netherlands

*Correspondence to:* Arve Kylling
(arve.kylling@nilu.no)

**Abstract.** Aerosol layer height is an essential parameter to understand the impact of aerosols on the climate system. As part of the European Space Agency Aerosol_cci project, aerosol layer height as derived from passive thermal and solar satellite sensors measurements, have been compared with aerosol layer heights estimated from CALIOP measurements. The Aerosol_cci project targeted dust type aerosol for this study. This ensures relatively unambiguous aerosol identification by the CALIOP pro-

cessing chain. Dust layer height was estimated from thermal IASI measurements by four different algorithms (BIRA-IASB, DLR, LMD, LISA) and from solar GOME-2 (KNMI) and SCIAMACHY (IUP) measurements. Due to differences in overpass time of the various satellites, a trajectory model was used to move the CALIOP derived dust heights in space and time to the IASI, GOME-2 and SCIAMACHY dust height pixels. It is not possible to construct a unique dust layer height from the CALIOP data. Thus two CALIOP derived layer heights were used: the cumulative extinction height defined to be the height

where the CALIOP extinction column is half of the total extinction column; and the geometric mean height which is defined as the geometrical mean of the top and bottom heights of the dust layer. In statistical average over all IASI data there is a general tendency to a positive bias of 0.5–0.8 km against CALIOP extinction-weighted height for three of the four algorithms assessed, while the fourth algorithm has almost no bias. When comparing to geometric mean height there is a shift by -0.5 km for all algorithms (getting close to zero for the three algorithms and turning negative for the fourth). The standard deviation

of all algorithms is quite similar and ranges between 1.0 and 1.4 km. When looking at different conditions (day, night, land, ocean) there is more detail in variabilities (e.g. all algorithms overestimate more at night than at day).



# 1 Introduction

Aerosol is identified as an essential climate variable (ECV) by the Global Climate Observing System (GCOS, http://www.wmo.int/pages/prog/gcos/). The aerosol layer height (GCOS product A.10.3) is one of four aerosol parameters which is needed to enhance our understanding of the aerosols' role in the climate system. Furthermore a deeper insight is important for radiative
budget analysis, study of chemical and physical interactions in the troposphere, weather forecast modelling, remote sensing and air quality initiatives. Ground-based methods (LIDARs) offer high accuracy and calibration benchmarks, however their geographical coverage is sparse. Hence, satellite observations of the aerosol layer height are warranted and the quality of such a product needs to be assessed. As part of the European Space Agency (ESA) Climate Change Initiative (CCI, Hollmann et al., 2013) the Aerosol_cci (Popp et al., 2016) project has conducted a comparison between dust type aerosol layer heights from
passive and active sensors to identify strengths and possible weaknesses in the estimate of this parameter.

Both active and passive methods may be used to estimate the aerosol layer height. The Cloud-Aerosol Lidar with Orthogonal Polarization (CALIOP) onboard the Cloud-Aerosol Lidar and Infrared Pathfinder Satellite Observations (CALIPSO) satellite (Winker et al., 2009, and https://www-calipso.larc.nasa.gov/) provides detailed vertical information with a vertical resolution of 30 m below 8.2 km and a horizontal footprint of 335 m. Passive solar and thermal infrared satellite instruments may provide
global data on a daily basis with horizontal resolution on the order of tens of km. For example Vandenbussche et al. (2013) retrieved desert dust aerosol vertical profiles from Infrared Atmospheric Sounding Interferometer (IASI) measurements; Cuesta et al. (2015) described the three-dimensional distribution, including dust height, of a dust outbreak over East Asia also using IASI measurements; Sanders and de Haan (2013) used the $O_2$ A-band to retrieve aerosol layer height from the Global Ozone Monitoring Experiment-2A (GOME). Dust top height may also be estimated using stereo view techniques by either utilizing
instruments with multi-angle capabilities (for example the Advanced Along Track Scanning Radiometer, AATSR, Virtanen et al., 2014) or by combining measurement from different sensors (see for example Merucci et al., 2016).

The aim of this work is to assess the different aerosol layer height products from different algorithms for various solar (SCanning Imaging Absorption SpectroMeter for Atmospheric CHartographY-SCIAMACHY, GOME-2) and thermal (IASI) sensors by comparison with CALIOP. The Aerosol_cci project targeted dust type aerosol for this study. The relatively unam-
biguous classification of dust by CALIOP and the availability of large dust events possibly avoid any biases due to aerosol misclassfication in the aerosol height comparison.

Earlier studies (Capelle et al., 2014; Peyridieu et al., 2010, 2013) have compared monthly averaged and gridded data, or a specific episode (Vandenbussche et al., 2013; Cuesta et al., 2015). We perform a point by point comparison for selected episodes. Furthermore we account for differences in satellite overpass time by the use of trajectory model analysis. Finally,
for the first time, utilizing data from GOME-2, SCIAMACHY and IASI with their respective spectral and spatial resolutions, results from the different passive infrared and solar algorithms are compared for the same dust episodes to identify strengths and weaknesses. Note that this work focuses on the comparison of dust layer heights retrieved from active and passive sensors. Comparisons of aerosol dust amount are outside the scope of this work and is discussed elsewhere (Popp et al., 2016).





The remainder of the paper is organized as follows: In Sect. 2 the data and data analysis methods are presented. The results from the aerosol layer height comparison are given in Sect. 3. The results are discussed in Sect. 4 and followed by the conclusions.

## 2 Data and methodology

To allow inclusion of data from the SCIAMACHY instrument that ceased operation in 2012, four desert dust events of Saharan origin in 2010 were selected (total 40 days):

- 18-27 March (10 days)

- 22 May - 1 June (11 days)

- 1-12 July (12 days)

- 14-20 September (7 days)

Furthermore we choose to focus the comparison on the region between 0-40°N and 80°W-120°E, see also upper plots of Figs. 2-7.

### 2.1 Active instrument dust height retrievals - CALIOP

CALIPSO is the fourth of the six satellites in the A-train satellite constellation. All six of the A-Train satellites cross the equator
within a few minutes of one another at around 13:30 local time. CALIOP is part of the payload of the CALIPSO platform. The CALIOP laser produces simultaneous coaligned pulses at 532 and 1064 nm that are used to measure the backscatter profile. The 532 nm pulse is linearly polarized. The return signal is polarized parallel and perpendicular to the outgoing plane and detected by two photomultiplier detectors. The CALIOP aerosol typing algorithm uses layer-averaged depolarization and the 532 nm attenuated backscatter to classify the aerosol into one of six types: clean marine, dust, polluted continental, clean
continental, polluted dust, smoke (Omar et al., 2009). Of the six types the dust aerosol is largely nonspherical implying a relatively large depolarization ratio and hence relatively unambiguous classification. Numerous data products are available from CALIOP. We use the 5 km profile product from CALIOP data version V4-10. Only dust profiles with CALIOP cloud aerosol discrimination (CAD) values between -100 and -20 are included (Winker et al., 2013). Profiles containing polluted dust and water and ice clouds are excluded. Furthermore we only include dust layers that are continuous, that is multi-layered
dust clouds are excluded from the analysis. The dust layer height is estimated from the extinction coefficient at 532 nm (`Extinction_Coefficient_532`). The extinction coefficient is a retrieved quantity and we only include profiles for which the quality control flag `Extinction_QC_Flag_532` equals 0 (unconstrained retrieval; initial lidar ratio unchanged during solution process) or 1 (constrained retrieval).





### 2.1.1 CALIOP dust layer height

Aerosol layers heights may be termed as either "effective" or "real" heights. The "effective" layer height represents the height where the total aerosol load should be placed in order to be representative for the radiative properties of this aerosol. The thickness of "effective" layers are typically assumed to be small, 500 m or 1 km. For climate impact studies "effective" layer

height is an important parameter, as it, together with the aerosol optical depth, single scattering albedo and phase function, allows quantitative estimates of the aerosols' direct radiative forcing. The "real" aerosol height may be described in terms of layer boundaries or by the full vertical profile. It is required for example for the understanding and characterising of aerosols-clouds interactions, air quality and flight safety.

There is no unique way to calculate the height of a dust layer from CALIOP data. Possible methods include:

**Threshold:** Calculate the cumulative extinction and set the height to be where cumulative extinction is above a prescribed threshold.

**Cumulative extinction:** Calculate the cumulative extinction and set height to be where the extinction column is half of the total extinction column.

**Geometric mean:** Identify the top and bottom heights of the dust layer and set dust height to the mean of the two.

**Extinction-weighted:** weigh the dust layer height $z_i$ for layer $i$ with an appropriate parameter and calculate the weighted average, for example using the extinction coefficient $\beta_i$ as in Koffi et al. (2012):

$$z_{\text{CALIOP}} = \frac{\sum \beta_i z_i}{\sum \beta_i} \tag{1}$$

In Fig. 1 the extinction-weighted and geometric mean CALIOP dust heights are plotted versus the cumulative extinction CALIOP dust height for the days and region under study. The extinction-weighted and cumulative extinction methods (right

plot) are fairly similar except for heights below about 2.5 km where the extinction-weighted method gives slightly larger heights. The geometric mean method (left plot) generally gives larger heights than the cumulative extinction method. The geometric mean method is purely geometrical. The cumulative extinction, threshold, and extinction-weighted methods use the profile extinction information from CALIOP. Below we present results for one CALIOP height method that includes extinction information and one that is purely geometrical. The extinction-weighted and cumulative extinction methods are nearly similar,

$R^2$=0.9942, right plot Fig. 1. To avoid having to arbitrarily set a threshold for the threshold method, we thus below present results for the cumulative extinction and the geometric mean methods.

It is noted that ambiguities in dust heights derived from CALIOP are larger for thick and optically dense dust layers. In these cases, the inversion of lidar profiles is less accurate for the lower part of these layers (due to uncertainties in the lidar ratio for dust) and also due to multiple scattering effects (see e.g. Cuesta et al., 2009, 2015). Multiple scattering effects are neglected in

the CALIOP operational products used here.





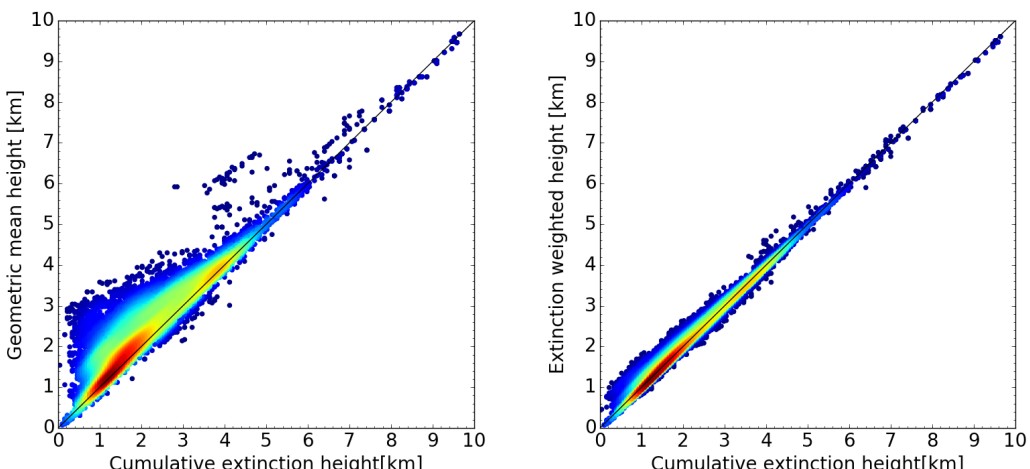

**Figure 1.** (Left) The CALIOP cumulative extinction height versus the CALIOP geometric mean height. Linear regression ($y = ax+b$) gives $a = 0.882$, $b = 0.6813$, $R^2$=0.941, and RMSE=0.652 km. (Right) The CALIOP cumulative extinction height versus the CALIOP extinction weighted height. Linear regression analysis gives, $a = 0.9619$, $b = 0.1597$, $R^2$=0.9942, and RMSE=0.182 km.

## 2.2 Passive instrument dust height retrievals

The dust layer height was estimated from measurements by IASI and GOME-2 onboard the MetOP-A satellite and SCIA-MACHY onboard Envisat. MetOP-A orbits in a sun-synchronous mid-morning orbit, crossing the equator at 9:30, local solar time, in the descending node. Envisat was in a sun-synchronous polar orbit crossing the equator at 10:00 AM local solar time
(MLST) in the descending node. The various dust height retrieval algorithms used in this study are summarized in Table 1 and described in more detail below.

### 2.2.1 The IASI algorithm at BIRA-IASB: MAPIR

The Mineral Aerosol Profiling from Thermal Infrared (MAPIR) retrieval algorithm is an extensive technical and scientific improvement of the algorithm first published by Vandenbussche et al. (2013). Versions 3.2 and 3.4 of the algorithm are used in
this study. The improved algorithm will be fully described in a separate publication, currently in writing.

The MAPIR retrieval scheme is based on the Optimal Estimation Method (OEM, Rodgers, 2000), which iteratively adjusts a state vector, composed of 7 variables: the surface temperature ($T_s$) and the vertical profile of dust aerosols concentration, from 1 to 6 km height in steps of 1 km. One of the strengths of the OEM is that it also produces averaging kernels, which enable the analysis of the sensitivity of the retrievals at the different heights, and the correlation between the retrieved parameters. The a
priori $T_s$ is, for retrievals using IASI data after 14 September 2010, taken from the IASI operational level 2 retrieval, version 5 and later (MAPIR version 3.2). For data prior to 14 September 2010, the $T_s$ retrieved by the IASI operational level 2 algorithm



**Table 1.** Summary of dust height retrieval algorithms. See text for further details, including description of acronyms.

| Institute | Radiative transfer | Satellite instrument | Algorithm specifics and aerosol type | Height | Reference(s) |
|---|---|---|---|---|---|
| BIRA-IASB | Line-by-line Lidort, Mie | IASI | Optimal estimation Dust, ash | Vertical profiles and averaging kernels; 1 km resolution from 1 to 6 km; 1.5 to 2 degrees of freedom; | Vandenbussche et al. (2013) |
| DLR | No direct forward modelling, optical properties from Mie | IASI | PCA, spectral matching, Bayesian probability Dust, ice clouds; possible application to ash | Effective layer height from emission temperature | Klüser et al. (2011), Klüser et al. (2012), Banks et al. (2013) |
| LMD | 4A/OP-DISORT | IASI | Refractive indices from Volz (1972, 1973) and Balkanski et al. (2007) | Average weighted layer height | Peyridieu et al. (2010), Peyridieu et al. (2013), Capelle et al. (2014) |
| LISA | Line-by-line KOPRA, Mie | IASI | Tikhonov-Philips auto-adaptive regularisation, Dust | Vertical profiles and averaging kernels; 1 km resolution from 0 to 9 km; approx. 1.5 degrees of freedom; | Cuesta et al. (2015) |
| KNMI | Line-by-line DISAMAR Henyey-Greenstein phase function | GOME-2 | Optimal Estimation All types | Effective layer height; accuracy of 0.5-1 km if AOD>0.3 | Sanders and de Haan (2013) |
| IUP | Line-by-line SCIATRAN T-matrix | SCIAMACHY | Adjoint RTE Dust | Effective layer height | Lelli et al. (2017) |



version 4 or prior have significant issues, in particular over deserts (giving unrealistic values). Therefore, for dust retrievals at those dates we use the $T_s$ from the European Centre Medium-Range Weather Forecast (ECMWF) era interim reanalysis as a priori (MAPIR version 3.4). The $T_s$ a priori standard deviation is set to 5 % in all retrievals. This might seem large but the diurnal variation of $T_s$ in deserts reaches more than 20 K.

As a priori for the vertical profile of desert dust concentration, we use the LIdar climatology of Vertical Aerosol Structure for space-based lidar simulation studies (LIVAS) monthly $1° \times 1°$ climatology derived from CALIOP data (Amiridis et al., 2015). That climatology contains mean vertical profiles of dust extinction at 1064 and 532 nm, with high vertical resolution. The extinctions in units of $km^{-1}$ have to be converted to vertical profiles of dust particles number concentration in units of $cm^{-3}$ at the vertical resolution of the retrieval. First, a linear interpolation is performed, from the CALIOP grid to the MAPIR grid. At

altitudes higher than 7 km, the aerosol concentration is set to zero. The extinction to concentration conversion is done using the visible (532 nm) cross-section of the aerosol particles used in MAPIR (using a single particle size, the median radius of the particle size distribution, PSD). A second issue arises here due to the fact that CALIOP measurements are sparse, and therefore the continuity of the climatology amongst adjacent $1° \times 1°$ cells is not ensured: the mean extinction in adjacent cells may come from measurements made for different days. To reduce this effect, we compute for the dust profile a priori a horizontal running

mean over 25 cells (5 in latitude, 5 in longitude). The standard deviation for the dust aerosol vertical profile is set to 100 % at all altitudes and all locations. When the surface altitude in a scene is higher than one (or more) retrieval altitude(s), the dust concentration at that (those) altitude(s) is set to zero prior to the retrieval (and can not deviate from this value during the retrieval).

  Three spectral windows in the thermal infrared are used for the retrievals, avoiding the huge ozone absorption band centered

at 1040 $cm^{-1}$ (9.6 μm): 905 to 927 $cm^{-1}$, 1098 to 1123 $cm^{-1}$ and 1202 to 1204 $cm^{-1}$. MAPIR relies on a very accurate radiative transfer code including multiple scattering: Lidort (Spurr et al., 2008). Radiative transfer calculations are performed on line for each retrieval. The forward modelling is computed as to properly reproduce the sampling and resolution of level 1c IASI data. The Lidort radiative transfer code is linked to a Mie code preparing the spherical aerosol optical properties from their PSD (log-normal, median radius 0.6 μm, geometric standard deviation of 2, corresponding to an effective size of 2 μm) and

refractive index (GEISA-HITRAN dust-like, Massie, 1994; Massie and Goldman, 2003; Jacquinet-Husson et al., 2011).

  Additional parameters are required for the radiative transfer calculations. The surface emissivity is taken from Newman et al. (2005) for oceans and from an updated version of the data reported in Zhou et al. (2011) for land. The two most important atmospheric profiles (temperature and water vapour) are taken from IASI level 2 operational products from EUMETSAT (procession version number depends on the date of the data set, the quality described in August et al. (2012)). Other relevant

atmospheric gas profiles (CO2, O3, N2O, CH4 and HNO3 are included) are taken from the US Airforce Geophysics Laboratory tropical climatology (Anderson et al., 1986). Line parameters of all gases come from the HITRAN 2012 database.

  Only scenes with less than 10 % cloud fraction from the IASI operational level 2 cloud product are retained for the dust retrievals. Unfortunately, the IASI cloud product seems to misflag some intense dust clouds as meteorological clouds, removing that data from our analysis.

After the retrievals, quality filters are undertaken. The retrievals are marked to be of good quality when:





- the root mean square of spectral residuals is lower than 2 K over land, 1 K over oceans

- the final fitted AOD at 10 μm is lower than 8 (otherwise the probability is extremely high that the scene was cloudy and unflagged as such).

The dust detection is computed a posteriori, and based on the single criterion that the retrieved 10 μm AOD must be higher than 0.01. This is a very low threshold, and although ensuring that all dusty scenes are indeed flagged as such, it might flag scenes where the aerosol presence is questionable. The AOD is obtained by vertical integration of the concentration profile, and multiplication by the extinction coefficient at the desired wavelength. The mean height is obtained from the profile as linear interpolation of the height that would separate the aerosol column in two identical partial columns (in other words, half the aerosols are below the mean height and half are above it).

### 2.2.2 IASI DLR algorithm - IMARS

The retrieval method for IASI developed at DLR combines dust and ice cloud remote sensing using Principal Components Analysis (PCA) of the high resolution IASI spectra (version 4.2). Bayesian inference is used for differentiating between dust and ice clouds (Klüser et al., 2011, 2012; Banks et al., 2013; Klüser et al., 2015). The method may also be applied to volcanic ash retrievals (Klüser et al., 2015; Maes et al., 2016), with focus on the very variable composition of the ash of which there is lack of reliable reports about in the literature.

The retrieval uses spectral pattern matching between 8 and 12 μm in a subspace of the observation space, spanned by suited eigenvectors for inferring dust/ash properties from the observations. With this approach direct forward modelling of the infrared radiative transfer is avoided, as this would be strongly underdetermined due to the lack of information about surface emissivity (over deserts), atmospheric temperature, humidity profiles as well as about detailed information of dust/ash composition, particle size and sphericity. The composition of dust/ash is assumed to be represented by linear combinations of typical dust composition mixtures (Klüser et al., 2015). For extreme cases not represented by these mixtures (e.g. very high calcite of gypsum content in dust aerosols) the retrieval will not be able to correctly characterize the dust/ash load. Dust optical properties used here have been calculated with traditional Mie theory, thus ignoring particle non-sphericity (see Klüser et al., 2015, 2016; Maes et al., 2016). One of the outputs of the DLR algorithm is the dust/ash emission temperature. Using a vertical temperature profile (standard atmosphere or model output) it is then converted to effective layer height (Klüser et al., 2015). De facto, the emission temperature is retrieved relative to the background, which implicitly delivers height information and not an absolute temperature.

The layer height is an effective emission height of a geometric thin (delta shape) dust layer. This makes its interpretation with regard to an averaged CALIPSO extinction profile non-intuitive. For optically thin dust layers the effective layer height is similar to the mean extinction of the profile; however, with growing dust AOD it moves further up in the profile (details depend on dust properties). Thus we expect a positive bias of the IMARS layer height to the cumulative extinction height from CALIPSO.



### 2.2.3   IASI LMD algorithm

The LMD method for the retrieval of dust characteristics from IASI observations was originally developed for application to the Atmospheric Infrared Sounder (AIRS) (Pierangelo et al., 2004, 2005), and then slightly modified as described in detail in Peyridieu et al. (2010, 2013) and in Capelle et al. (2014) for application to IASI. The method used to derive dust characteristics

from IASI observations is a three-step physical algorithm based on a "Look-up Table" (LUT) approach. The first step constrains the atmospheric state (temperature and water profile) using 18 channels selected in the spectral range 4.5-14.5 µm and mostly sensitive to temperature and water profiles between 900 hPa and 200 hPa and not, or almost not, sensitive to surface characteristics (temperature, emissivity). The second step determines simultaneously the 10 µm AOD, the dust layer mean height and the surface temperature using 8 channels localized in three window regions: 8-9 µm, 10-12 µm and 4.6-4.7 µm.

This selection of channels, both at short and long wavelengths, is aimed at decorrelating the contribution of AOD, height and surface temperature to the observed signal. The dust coarse mode particle effective radius can be determined in a third step.

For each step, LUTs of IASI simulated brightness temperatures are calculated using the forward coupled radiative transfer model 4A/OP-DISORT (available from http://4aop.noveltis.com, Scott and Chédin, 1981). Entries to the model include: AOD, height, surface pressure, surface temperature and emissivity, viewing angle, two refractive indices: Volz (1972, 1973) and

Balkanski et al. (2007), and a set of 2311 atmospheric situations, selected by statistical methods from 80,000 radiosonde reports, and stored in the Thermodynamic Initial Guess Retrieval (TIGR) climatological database (Chédin et al., 1985; Chevallier et al., 1998). The PSD is modeled by a monomodal lognormal distribution described by the effective radius ($R_{\mathrm{eff}}$) and the standard deviation of the distribution $\sigma_g$. Following results of previous sensitivity studies (Appendix A, Capelle et al., 2014; Pierangelo et al., 2005), fixed values are taken for the effective radius ($R_{\mathrm{eff}} = 2.3$ µm) and for the standard deviation of the size distribution

($\sigma_g = 0.65$).

The aerosol vertical distribution is supposed to be concentrated within a single homogeneous layer. While this assumption cannot describe correctly observations, in general more complex, the height retrieved here can be defined as an average weighted height for which half of the dust optical depth is below and half of the optical depth is above. This infrared optical equivalent to the real vertical profile is therefore appropriate for computing dust infrared forcing. It is worth noting that

the resulting mean layer height corresponds to height above sea level. Several aspects of the retrieval algorithm: robustness to aerosol model (size distribution, shape, and refractive indices), possible contamination by other aerosol species, radiative transfer model bias removal, or cloud mask including discrimination between clouds and aerosols, etc., were investigated and details may be found, for example, in Pierangelo et al. (2004) and Capelle et al. (2014). The surface emissivity spectrum is supposed to be known and is read from a 0.5° monthly grid retrieved from IASI (Capelle et al., 2012).

### 2.2.4   IASI LISA algorithm - AEROIASI

The IASI algorithm from LISA, called AEROIASI, has been conceived to observe the three-dimensional (3D) distribution of desert dust plumes for each overpass of IASI, both over land and ocean (Cuesta et al., 2015). For this, it derives vertical profiles of desert dust, in terms of the extinction coefficient at 10 µm, from individual thermal infrared spectra measured by IASI. It





is a constrained-least-squares fit method, based on explicit radiative transfer calculations, where the vertical distribution and abundance of dust are iteratively adjusted in order to fit IASI observations. This approach uses auto-adaptive constraints for adjusting simultaneously the dust profile and surface temperature in order to offer particularly good adaptability for different atmospheric and surface conditions. This flexibility makes the aerosol retrieval possible for most cloud-free IASI pixels, both

over ocean and land (even for bright surfaces and relatively low aerosol loads). The information on the vertical distribution of dust is mainly provided by their broadband radiative effect, which includes aerosol thermal emission depending at each height on the vertical profile of temperature (assuming local thermal equilibrium).

AEROIASI uses an a priori desert dust model (including dust microphysical properties) and meteorological profiles provided as inputs to the radiative transfer model. The line-by-line Karlsruhe Optimized and Precise Radiative transfer Algorithm

(KOPRA, Stiller, 2000) is used to simulate thermal infrared radiance spectra and the inversion module KOPRAFIT to compare them to those measured by IASI, for 12 selected spectral micro-windows in the atmospheric window between 8 and 12 μm. KOPRA accounts for light absorption, emission, and single scattering by aerosols, using dust optical properties derived at each wavelength with a Mie code (Metzig, 1984) optimized as described in Deirmendjian et al. (1961) and Kerker (1969). The vertical grid of all profiles in the simulations is set between the surface and 9 km height asl (above mean sea level), with 1 km

increments. For each pixel, we use atmospheric temperature profiles and first guesses of surface temperatures and water vapor profiles from ERA-Interim (ERAI) reanalysis (Dee et al., 2011) of the ECMWF. For all seasons and locations, AEROIASI uses a unique a priori vertical profile of dust derived from CALIOP average dust profiles over the Sahara on the summer of 2011.

In order to minimize the spectral residuals, the method adjusts iteratively the radiative transfer inputs (mainly the aerosol vertical profile and surface temperature) until reaching convergence (the maximum number of iterations is fixed to 10). The re-

trieved dust profile is obtained with an auto-adaptive Tikhonov-Phillips-type (Tikhonov, 2003) height-dependent regularization conceived to avoid unrealistic oscillations in the retrieved dust profiles while adapting the results to the amount of information provided by the IASI measurements. The overall results of AEROIASI show that the degrees of freedom or the number of independent pieces of information in the retrieval of dust profiles varies during the iterative procedure, a typical value of ~1.5 being used to determine the shape of the dust extinction profiles.

Once IASI spectra are fitted, a series of quality checks are performed in order to screen out cloudy measurements and aberrant retrievals. We exclude IASI pixels with derived surface temperatures below their ERAI reanalyses counterparts by more than 10 K and those pixels exhibiting too high root-mean-squared spectral residuals or horizontal variability with respect to their closest pixels. For each quality-checked retrieval, we derive a vertical profile of dust extinction coefficient ($\alpha_{10}$ in $km^{-1}$) at 10 μm, the associated AOD (by vertical integration of the extinction profile), and mean and top heights of the observed dust

layer. The mean height of the observed dust layer (the product used in the current paper) is defined as the height below which the integral of the extinction coefficient profile reaches 50 % of the AOD. Likewise, the height of the top of the dust layer is defined as the height below which the integral of the extinction coefficient profile reaches 95 % of the AOD. Indeed, AEROIASI provides valuable information not only on the mean height of the dust layers but also on their vertical extent (i.e., layer top heights and whether the layers reach the ground or are elevated.



In this study AEROIASI retrievals from version 2 of the algorithm are used. It mainly differs from the previous version described by Cuesta et al. (2015) in the a priori desert dust model and the surface emissivity database. Using these new databases, we obtain lower spectral residuals with respect to IASI measurements than with the previous version and higher adaptability for covering the large region analysed in this paper (i.e. the tropical dust belt). The climatological desert dust

model consists of refractive indices, a single-mode lognormal particle size distribution and an a priori vertical profile of dust. Refractive indices are taken from field measurements of Saharan dust analysed by Di Biagio et al. (2014). The modal radius and width of the single-mode distribution are prescribed from average volume effective radius and width for the coarse mode (for radii>0.6 μm of the AERONET size distributions, http://aeronet.gsfc.nasa.gov, Dubovik et al., 2002) derived from Saharan ground-based stations in June 2011. A unique first guess of dust vertical distribution (the same profile for all pixels and all

seasons) is considered in the inversion, which is obtained from an average of CALIOP extinction vertical profiles for dust over the Sahara (during large dust outbreaks in late June 2011), scaled to particle concentration units (in order to set an a priori AOD at 10 μm of 0.03). Forward simulations include surface emissivity from a global monthly IASI-derived climatology over land (Paul et al., 2012) and a surface temperature dependent model over ocean (Newman et al., 2005).

### 2.2.5   GOME-2 KNMI algorithm

The deep oxygen lines (A band and/or B band) in the near infrared of the shortwave spectrum traditionally have been used for retrieval of the cloud height. In the absence of clouds, these bands contain information on the aerosol height (Wang et al., 2012). The algorithm developed at KNMI within the TROPOMI/Sentinel 5 Precursor programme (Veefkind et al., 2012) is based on the optimal estimation method and aims at deriving the aerosol layer height (Sanders and de Haan, 2013). This method has also been applied to Greenhouse gases Observing SATellite (GOSAT) and GOME-2 data within the on-going ESA AEROPRO

study, in support of the Sentinel-4 development. The algorithm is sensitive to all aerosol types, including dust, biomass burning and industrial pollution plumes. Sensitivity analyses performed for the TROPOMI/Sentinel 5 Precursor ATBD indicate that the aerosol layer height can be derived with an accuracy of 0.5-1 km, if the AOD is 0.3 or larger.

The algorithm uses the Determining Instrument Specifications and Analyzing Methods for Atmospheric Retrieval (DISAMAR) retrieval and simulation package developed at KNMI. In the setup that is used in this work, the aerosol is modelled as a 50 hPa

thick layer, for which the height and the aerosol optical depth is fitted. The single scattering albedo of the aerosol particles is assumed to 0.95 and we apply a Henyey-Greenstein phase function with an asymmetry parameter of 0.7. A climatological value for the surface reflectance is used. Pressure-temperature profiles are obtained from the operational ECMWF forecast. The algorithm uses a fit window between 758 and 762 nm.

The algorithm is only applied to cloud-cleared scenes for which the UV aerosol index has a value exceeding 1.0, indicating

the presence of absorbing aerosol layers. For the GOME-2 data used in this work, the cloud clearing is done based on the GOME-2 data itself, which may result in undetected sub-pixel cloudiness. It is noted that the size of the GOME-2 ground pixels is much larger than for the TROPOMI instrument, for which the algorithm has been designed.



### 2.2.6 SCIAMACHY IUP algorithm

The height of an aerosol layer is determined using top-of-atmosphere (TOA) reflectances $R$ (defined as the sun-normalized radiances, weighted by the cosine of the solar zenith angle) in the oxygen A-band, that is, in the range 758-772 nm, at the nominal spectral sampling 0.21 nm of SCIAMACHY, for a Gaussian instrument response function of 0.48 nm. The retrieval is based on the calculation of the weighting functions $W(h, b, \tau) = \partial R(h, b, \tau)/\partial(h, b, \tau)$, i.e., the Jacobians of $R$ as function of the top and bottom altitude $h, b$ and optical thickness $\tau$ of the aerosol layer. Upon linearization of the problem, the measured $R$ in a gaseous absorption band can be written as a function of the desired $h$. Given that $\tau$ is inferred from an independent source, such as a non-absorbing channel outside the oxygen A-band, typically $\lambda = 758$ nm, the retrieval can be further simplified assuming either that the aerosol layer originates at the ground ($b = 0$ km) or is elevated ($b \neq 0$ km). The latter assumption implies that the prior geometrical thickness is preserved when retrieving $h$. Either way, the problem is reduced to the calculation of $W(h)$ (Rozanov, 2006; Rozanov et al., 2007) and the minimization of the difference between the forward-modelled and the measured reflectance, converging after $\sim$4 iterations on average, delivers the height of the layer.

Information on the local non-spherical dust optical properties, encoded in the spectral scattering T-matrix (Dubovik et al., 2006), as well as the single-scattering albedo and the aerosol extinction (box) profiles are embedded in $W(h)$. It has been assumed that these quantities are independent of height inside the aerosol layer. The HITRAN 2008 edition (Rothman et al., 2009) is used for the line intensities of the absorbing species (oxygen and water vapour) included in the forward problem. The full retrieval chain is implemented and carried out within the radiative transfer model SCIATRAN (Rozanov et al., 2014)

The selection of cloud-free SCIAMACHY pixels ($60 \times 40$ km$^2$ of nominal footprint size) relies on the analysis of joint histograms of geometric cloud cover (CC < 0.1, from colocated $1 \times 1$ km$^2$ MERIS observations, Schlundt et al., 2011) and aerosol absorbing index (AAI > 1.0, de Graaf et al., 2005)) for the area of interest. Surface reflectivity is taken from the MERIS-derived black-sky dataset (Popp et al., 2011) which is the critical parameter for the accuracy of the retrieved $h$. Already an error of $\pm 10$ % in the a priori value of surface reflectivity can cause a bias up to $\pm 1$ km for $\tau = 0.25$ and $h > 3.0$ km. More details about the IUP algorithm, as well as validation with independent measurements when applied to an elevated ash layer, are given in Lelli et al. (2017).

### 2.3 Data selection and comparison methodology

The selection of data and the comparison between CALIOP and the other satellite instrument estimates of dust heights proceed through the following steps for each date listed above:

1. Identify CALIOP swaths that are within the region of interest.

2. Identify closest CALIOP swath and IASI, GOME-2 and SCIAMACHY dust pixels in time and space. Due to the difference in the CALIPSO equator crossing time (13:30) and MetOP-A and Envisat equator crossing times (09:30 and 10:00), a maximum time difference of 5 hrs is allowed between CALIOP and IASI, GOME-2 and SCIAMACHY dust pixels in this step. To allow for possible movement of dust pixels between overpasses, pixels within 500 km were included for subsequent analysis. This allows for a maximum wind speed of 100 km/hr.





3. For the CALIOP swaths from step 2 identify CALIOP dust profiles using the CALIOP dust flag and CAD score. Calculate CALIOP cumulative extinction and geometric mean dust layer heights.

4. Move CALIOP dust layer heights from the previous step backward in time to the Metop-A and Envisat overpass times using the FLEXTRA trajectory model.

5. After moving the CALIOP dust heights backward in time they may still be at locations different from the IASI, GOME-2 and SCIAMACHY dust heights. A second colocation is thus made to colocate the moved CALIOP dust heights with IASI, GOME-2 and SCIAMACHY dust heights. The maximum difference in distance is set to 20 km for IASI and 100 km for SCIAMACHY and GOME-2 reflecting the larger footprints of the latter two instruments.

6. Analysis of height differences including statistics.

In the top panels of Figs. 2-7 are shown examples of data from steps 1-5. The pixels identified as dust from IASI data by the BIRA-IASB (Fig. 2), DLR (Fig. 3), LMD (Fig. 4), and LISA (Fig. 5) algorithms, the KNMI GOME-2 (Fig. 6) and the IUP SCIAMACHY dust data (Fig. 7) are overlaid by CALIOP cumulative extinction heights, derived from profiles identified as dust (step 3, grey dots), that are within the temporal and spatial requirements. CALIOP data are recorded after the IASI overpass. To account for possible movements of the dust between the overpasses the CALIOP dust heights were moved in longitude, latitude and height using the FLEXTRA model (step 4, Stohl et al., 1995; Stohl and Seibert, 1998). FLEXTRA calculates mean-wind trajectories using data from the ECMWF. It does not include turbulence or loss processes. The blue dots in the upper plots of Figs. 2-7 are CALIOP dust height pixels that have been moved from their original location (grey dots) to the nearest IASI pixel (step 5). As the cumulative extinction and geometric mean CALIOP dust heights are different they will be moved by FLEXTRA to different locations. This is seen in the curtain plots in Figs. 2-7 where the cumulative extinction (purple dots) and geometric mean (pink triangles) heights from the passive instruments sometimes overlap (not moved or moved to same location and height) and sometimes do not overlap (moved to different location and/or height). It is also seen in the difference in number of colocated points, compare grey and light grey entries in Table 2. For the full data period the CALIOP dust heights were on average moved upwards by 0.015 (cumulative extinction) and 0.020 km (geometric mean), both with standard deviations of 0.25 km.

# 3 Results

The analysis steps 1-5, section 2.3, were performed for all days and algorithms. The number of dust pixels identified by the various algorithms after step 1 is given in Table 2. The number of pixels identified as dust by the various IASI-algorithms varies by a factor of 4.6. The differences reflect the differences in dust detection methods and it is outside the scope of this study to further investigate the reasons for these differences. As expected the solar algorithms detect far fewer dust pixels due to only day time coverage (factor of 2) and larger pixels size (factor of about 16). The difference between the two solar algorithms (KNMI and IUP) are due to differences in the constraints set to detect dust. In step 2 dust pixels within a given time and distance from the CALIOP detected dust pixels are selected. This step reduces the number of IASI data points to





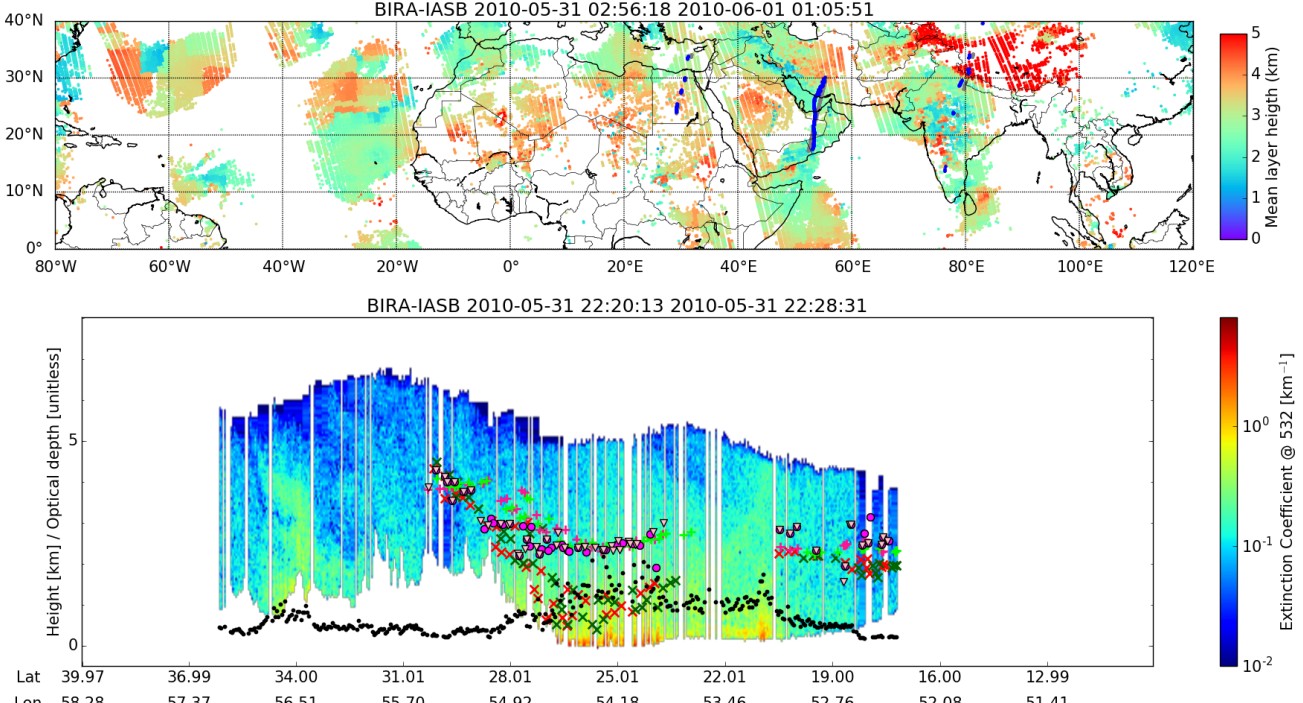

**Figure 2.** (Top) The IASI dust height from the BIRA-IASB analysis. The CALIOP profiles identified as dust, within 500 km and 5 hrs time differences from nearest IASI pixel, are overlaid (grey dots). The location of the CALIOP height after shifting to IASI overpass time is shown by the blue dots. The time range (UTC) in the title gives the times of the first and last CALIOP points plotted. (Bottom) Curtain plot of the CALIOP extinction coeffcient for heights identified as dust. IASI-BIRA-IASB dust layer heights are shown as purple dots (colocated with CALIOP cumulative extinction heights) and pink triangles (colocated with CALIOP geometric mean heights). Red crosses are CALIOP cumulative extinction heights that have been shifted to the IASI pixel location using FLEXTRA. The green crosses are the CALIOP cumulative extinction heights before the shift. Deep pink plusses are CALIOP geometric mean heights that have been shifted to the IASI pixel location using FLEXTRA. The lime plusses are the CALIOP geometric mean heights before the shift. The black dots are the column optical depth at 532 nm from CALIOP. The curtain is for the CALIOP data between 40-60°E in the top plot. The time range (UTC) in the title gives the times of the first and last CALIOP extinction coefficient profiles plotted. See text for further details.

between 0.58-1.8 % of those in step 1. The number of GOME-2 and SCIAMACHY points are reduced to 17.3 and 73.0 %, respectively. The movement of CALIOP dust heights to the MetOp-A and Envisat overpass times and the final colocation of CALIOP heights and dust pixels give the final number of dust heights to be compared to CALIOP dust heights, see values for step 5 in Table 2, and grey and light grey entries in Table 3.

5    Inspection of IASI retrieved dust heights shown in the upper plots of Figs. 2-5 reveal differences in dust detection and dust height between the various algorithms. While differences in dust detection is not the subject of this paper, we do, however, note



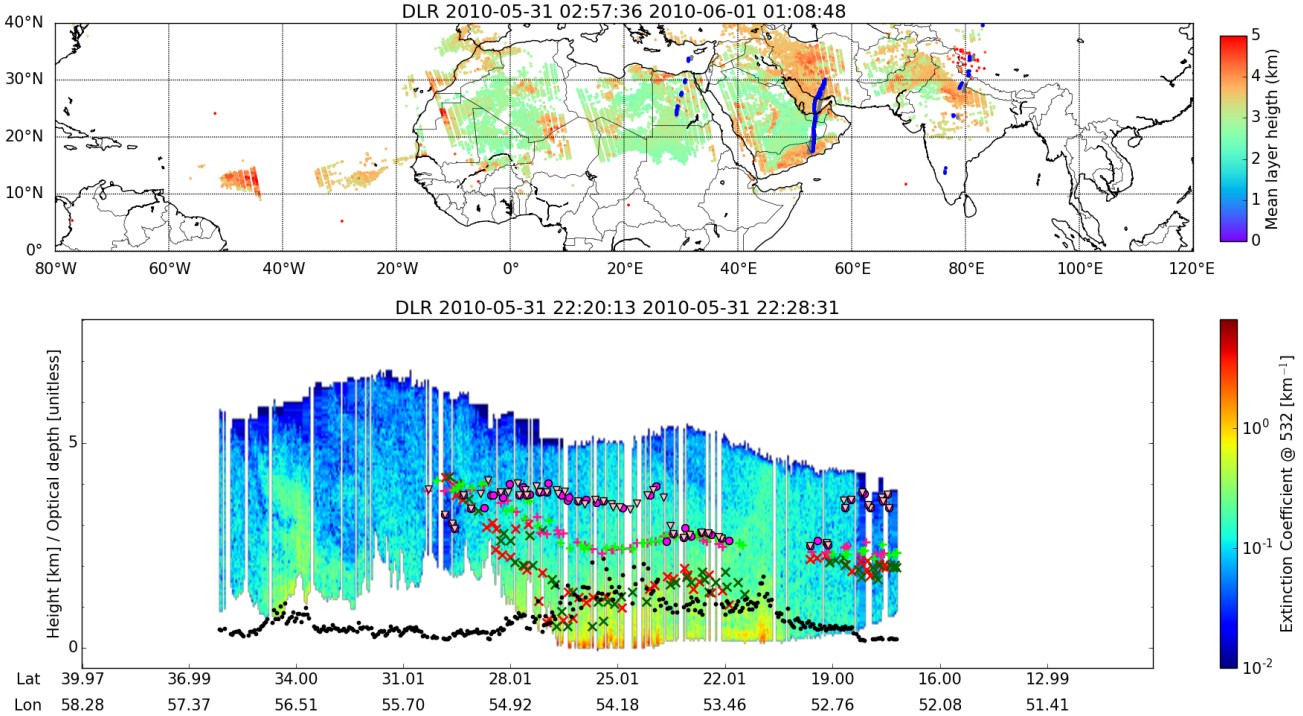

**Figure 3.** Similar to Fig. 2, but IASI dust height from DLR analysis.

**Table 2.** The number of data points (dust heights) at data reducing step of the data analysis chain described in section 2.3. Numbers in parenthesis are in percentage relative to the total number in the previous analysis step. Step number refers to the analysis steps as described in section 2.3.

| Step | BIRA-IASB | DLR | LMD | LISA | KNMI | IUP |
|---|---|---|---|---|---|---|
| 1 | 2324277 | 503944 | 811360 | 1770793 | 21535 | 2710 |
| 2 | 13377 (0.58) | 5208 (1.0) | 14916 (1.8) | 13110 (0.74) | 3715 (17.3) | 1979 (73.0) |
| 5-cumulative extinction | 2620 (19.6) | 1420 (27.3) | 748 (5.0) | 2203 (16.8) | 215 (5.8) | 34 (1.7) |
| 5-geometric mean | 2408 (18.0) | 1296 (24.9) | 704 (4.7) | 1978 (15.1) | 91 (2.4) | 21 (1.1) |

that there are substantial differences in the pixels identified as containing dust by the various algorithms. In particular the DLR algorithm detects very little dust over the ocean regions; the BIRA-IASB and LISA algorithms detect dust over the ocean west of 40°W and north of 20°N whereas the DLR and LMD algorithms do not detect dust in this region; for the example curtain plots in Figs. 2-5 all algorithms except LMD detects dust north of about 28°N. The BIRA-IASB-algorithm's detection of dust

5  over Himalaya is due to retrievals being undertaken for all non-cloudy scenes, and the final result may never be a true zero





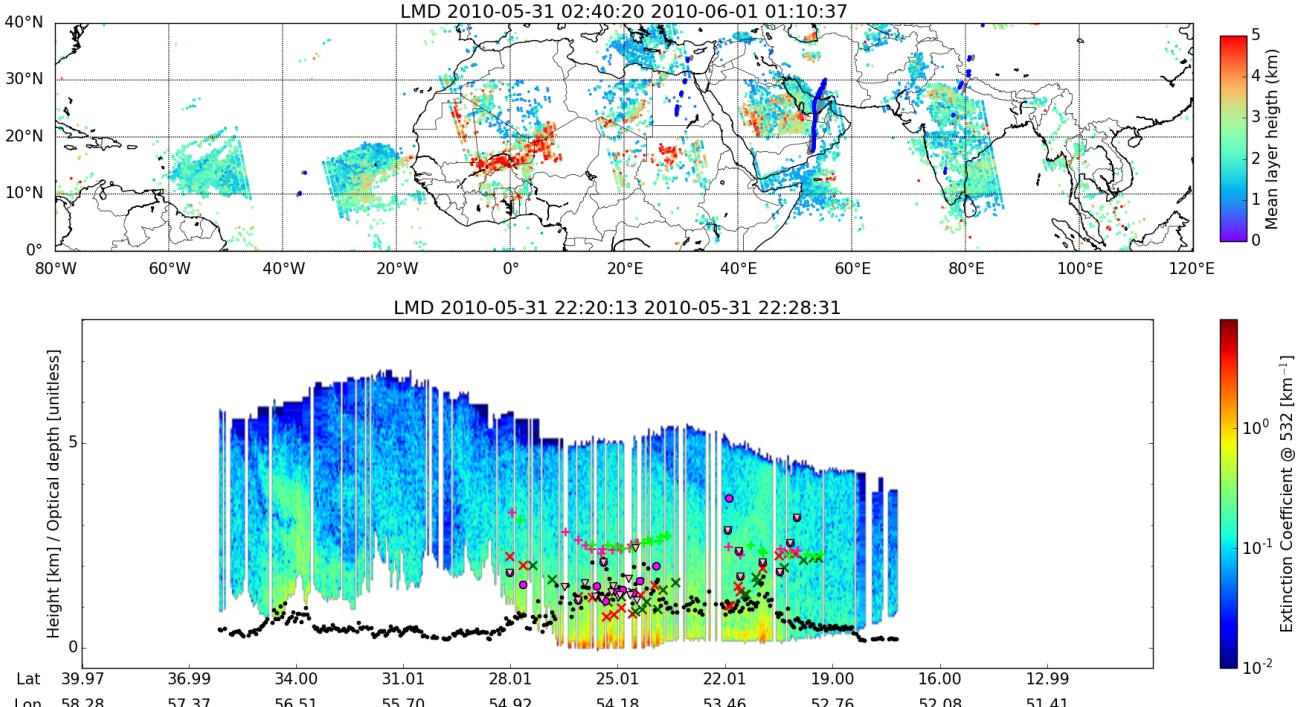

**Figure 4.** Similar to Fig. 2, but IASI dust height from LMD analysis.

due to the method used. These retrievals have a low aerosol optical depth (AOD) and their inclusion indicates that the AOD threshold for the dust flag may be be too permissive. The differences in dust detection is the reason for the different number of pixels available for comparison with CALIOP.

The curtain plots in Figs. 2-5 show that the BIRA-IASB-algorithm (purple dots and pink triangles) agrees reasonably with

5    the CALIOP geometric mean heights (deep pink plusses) and gives higher dust heights compared to the CALIOP cumulative extinction heights (red crosses). For the DLR-algorithm the situation is similar, albeit the DLR-algorithm generally gives larger dust heights. The LMD-algorithm heights are generally similar to CALIOP cumulative extinction heights while LISA-algorithm heights are in better agreement with the geometric mean heights. For this transect, BIRA-IASB and LISA algorithms capture the rather monotonous decrease of dust layers heights from about 4 km of altitude near 30°N to 2 km of altitude at 19°N

10    depicted by CALIOP geometric mean heights. The LMD algorithm retrieves dust heights near 1.5 km of height at 24-27°N as for CALIOP cumulated extinction estimates. The behaviour for this single overpass is also present in the full IASI data set as shown in Figs. 8-9 and Table 3. Do, however, note that there are substantial differences when comparing the passive methods with the CALIOP cumulative extinction and geometric mean methods. Overall, the CALIOP geometric mean method gives a larger CALIOP dust height. Thus, the CALIOP minus passive instruments difference is smaller for the geometric mean method



**Table 3.** The mean ( yellow , green and blue boxes) of the dust height difference between the passive sensors and CALIOP. Also given is the standard deviation ($\sigma$) and number (#) of colocated points. The inlay ( apricot ) is the percentage of heights that are within the CALIOP layer. Bright colored numbers are for the cumulative extinction heights while the dimmed colors are for the geometric mean CALIOP dust heights. For BIRA-IASB, DLR, LISA and LMD statistics are given for all ( yellow ) data and subgroups of data recorded during day and night and over land ( green ) and ocean ( blue ). For KNMI-GOME2 only day comparisons are possible, hence the lack of comparisons with CALIOP night overpasses. Note that for KNMI the number of land and ocean pixels does not add up to the total due to some pixels covering coastal regions (mixed pixels).

| | BIRA-IASB mean (km) / points (#) | $\sigma$ / inlay (%) | DLR mean (km) / points (#) | $\sigma$ / inlay (%) | LMD mean (km) / points (#) | $\sigma$ / inlay (%) | LISA mean (km) / points (#) | $\sigma$ / inlay (%) | KNMI mean (km) / points (#) | $\sigma$ / inlay (%) | IUP mean (km) / points (#) | $\sigma$ / inlay (%) |
|---|---|---|---|---|---|---|---|---|---|---|---|---|
| **All** | 0.590 | 1.213 | 0.785 | 1.281 | -0.053 | 1.339 | 0.507 | 1.126 | -0.818 | 1.455 | -0.961 | 1.708 |
| | 0.078 | 1.108 | 0.243 | 1.181 | -0.607 | 1.187 | -0.045 | 1.029 | -1.393 | 1.204 | -1.097 | 1.574 |
| | 2620 | 21.8 | 1420 | 15.1 | 748 | 32.5 | 2203 | 20.0 | 215 | 42.3 | 34 | 17.1 |
| | 2408 | 37.2 | 1296 | 28.8 | 704 | 52.4 | 1978 | 39.3 | 91 | 54.9 | 21 | 31.8 |
| **CALIOP Day** | | | | | | | | | | | | |
| **Land** | 0.357 | 1.665 | 0.405 | 1.660 | -0.102 | 1.448 | -0.225 | 1.454 | -0.229 | 1.339 | | |
| | 0.087 | 1.572 | -0.044 | 1.526 | -0.496 | 1.322 | -0.635 | 1.357 | -0.893 | 0.930 | | |
| | 605 | 18.2 | 377 | 10.6 | 319 | 31.7 | 440 | 27.7 | 117 | 33.3 | | |
| | 598 | 23.8 | 393 | 26.0 | 322 | 43.8 | 425 | 47.3 | 50 | 56.0 | | |
| **Ocean** | 0.783 | 0.913 | 0.913 | 1.539 | -0.501 | 1.409 | 0.172 | 1.389 | -1.477 | 1.296 | | |
| | 0.340 | 1.187 | 0.184 | 1.174 | -0.922 | 1.142 | -0.285 | 1.187 | -2.015 | 1.262 | | |
| | 172 | 26.7 | 22 | 31.8 | 118 | 17.8 | 180 | 10.6 | 85 | 56.5 | | |
| | 170 | 35.3 | 22 | 54.5 | 109 | 39.4 | 170 | 18.8 | 34 | 58.8 | | |
| **CALIOP Night** | | | | | | | | | | | | |
| **Land** | 0.567 | 1.020 | 0.906 | 1.062 | 0.073 | 1.092 | 0.663 | 0.896 | | | | |
| | 0.038 | 0.903 | 0.358 | 0.964 | -0.579 | 1.058 | 0.170 | 0.855 | | | | |
| | 1501 | 26.1 | 996 | 16.2 | 206 | 48.5 | 1226 | 22.9 | | | | |
| | 1330 | 42.6 | 854 | 29.9 | 177 | 68.9 | 1064 | 41.1 | | | | |
| **Ocean** | 1.008 | 0.741 | 1.599 | 1.127 | 0.352 | 1.180 | 1.043 | 0.637 | | | | |
| | 0.094 | 0.678 | 0.835 | 0.720 | -0.674 | 0.878 | 0.152 | 0.486 | | | | |
| | 342 | 6.7 | 25 | 24.0 | 105 | 20.0 | 357 | 5.3 | | | | |
| | 310 | 40.3 | 27 | 14.8 | 95 | 65.6 | 319 | 33.9 | | | | |





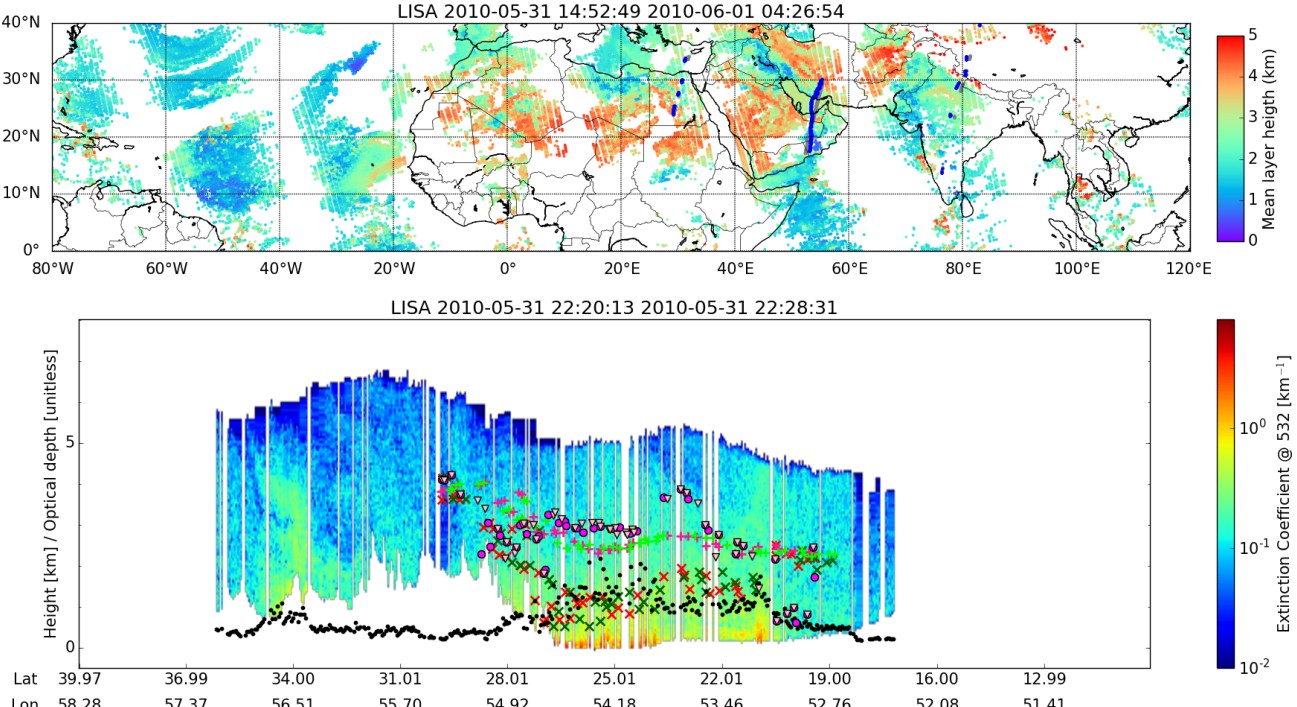

**Figure 5.** Similar to Fig. 2, but IASI dust height from LISA analysis.

compared to the cumulative extinction method. The geometric mean method also gives slightly smaller standard deviations and more dust heights from the passive instrument within the CALIOP dust layer, see the apricot colored entries in Table 3.

In the upper row of Fig. 8 the CALIOP cumulative extinction height is plotted against the dust heights from all IASI algorithms for all dates. Fig. 9 is similar but for the CALIOP geometric mean height method. Also included in the plots are the

5 Pearson's correlation coefficient and root mean square error (RMSE). In the centre rows of Figs. 8-9 the differences between the passive algorithms and CALIOP heights are shown versus the CALIOP column extinction. In the upper and centre rows the color indicates the density of points. In the bottom rows of Figs. 8-9 are shown the frequency distribution of the difference between the dust heights from the various IASI algorithms and the CALIOP heights. Also included is a fit to the normal distribution and the mean and standard deviation ($\sigma$) of the fit. Similar plots for the KNMI and IUP algorithms are shown in

Fig. 10. For the IASI algorithms ocean-day, ocean-night, land-day and land-night data subsets are presented in Figs. A1-A4 for the CALIOP geometric mean heights, and in Figs. A5-A8 for the CALIOP cumulative extinction heights The mean and standard deviation and the number of data points are also listed in Table 3. It is noted that an analysis in terms of "bias" is correct as a mean analysis only when the difference distribution is at least symmetrical (if not Gaussian). This is not always the case as shown for example for the ocean day subset in Fig. A8. Thus, while the mean of the difference may appear good

the histogram sometimes shows something very different.





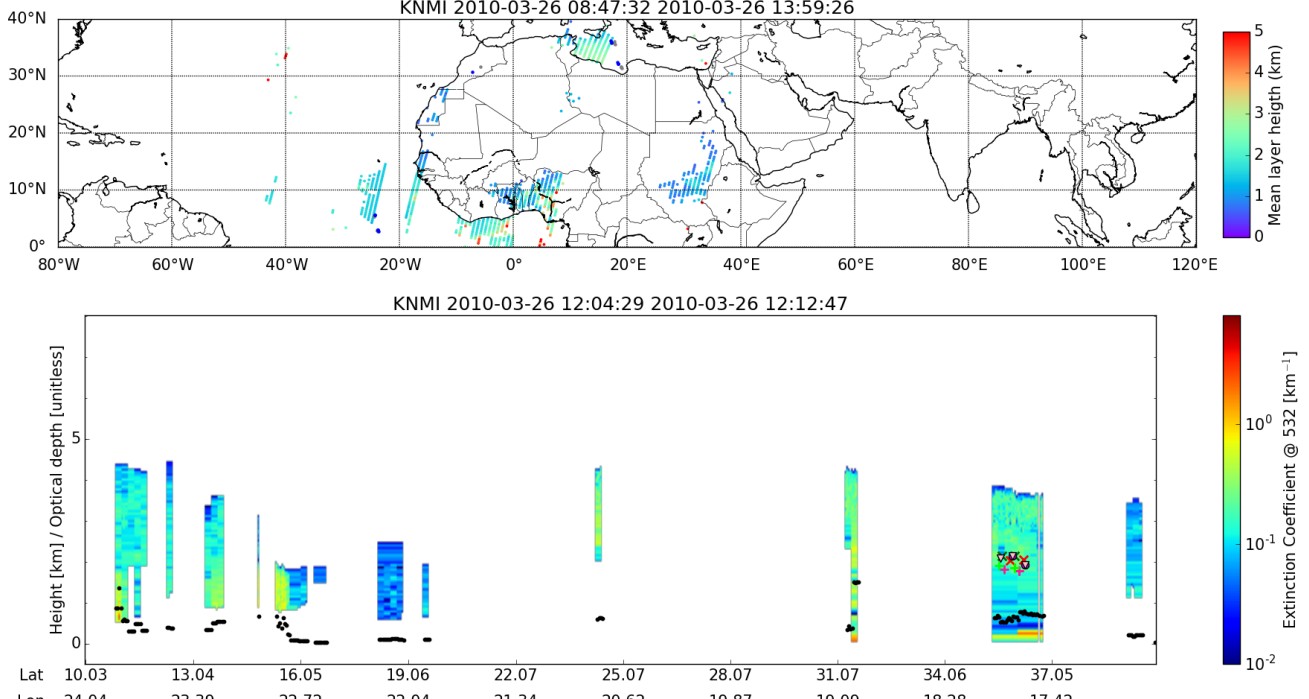

**Figure 6.** Similar to Fig. 2, but GOME-2 dust height from KNMI analysis. Note difference in time from Fig. 2. The curtain in the bottom plot is for the CALIOP data between 0-20°E in top plot.

For the BIRA-IASB, LISA and LMD algorithms versus the CALIOP dust cumulative extinction (geometric mean) height, the Pearson's correlation coefficient is between 0.408-0.510 (0.414-0.518) . It is smaller for the DLR, KNMI and IUP algorithms, being -0.115-0.120 (-0.238-0.137). For the IASI-algorithms the RMSE is between 1.030-1.334 km when comparing with the CALIOP geometric mean dust heights. It increases to 1.235-1503 km for the CALIOP cumulative extinction dust heights. For the KNMI and IUP algorithms the RMSE is larger, 1.670-3.439 km. The rather large RMSE indicates the difficulty and uncertainty involved when comparing dust heights from very different sensors and data recorded at different times with large differences in footprint size. There appears to be no dependence of height differences on dust column extinction as shown in the centre rows of Figs. 8-10.

For the IASI algorithms both day and night data are included in Figs. 8-9. The mean height difference between the various algorithms and the CALIOP heights are high-lighted in dimmed yellow (geometric mean) and bright yellow (cumulative extinction) in Table 3. The BIRA-IASB mean height difference is 0.078 km (0.590 km) when compared with the CALIOP geometric mean (cumulative extinction) height. The DLR algorithm mean height difference of 0.243 km (0.785 km) is larger. However, it is noted that the DLR algorithm generally gives the altitude at two distinct modes, Fig. 8. For LMD the magnitude of the mean height difference is smallest when comparing with the CALIOP cumulative extinction height, -0.053 km. It

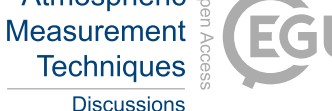



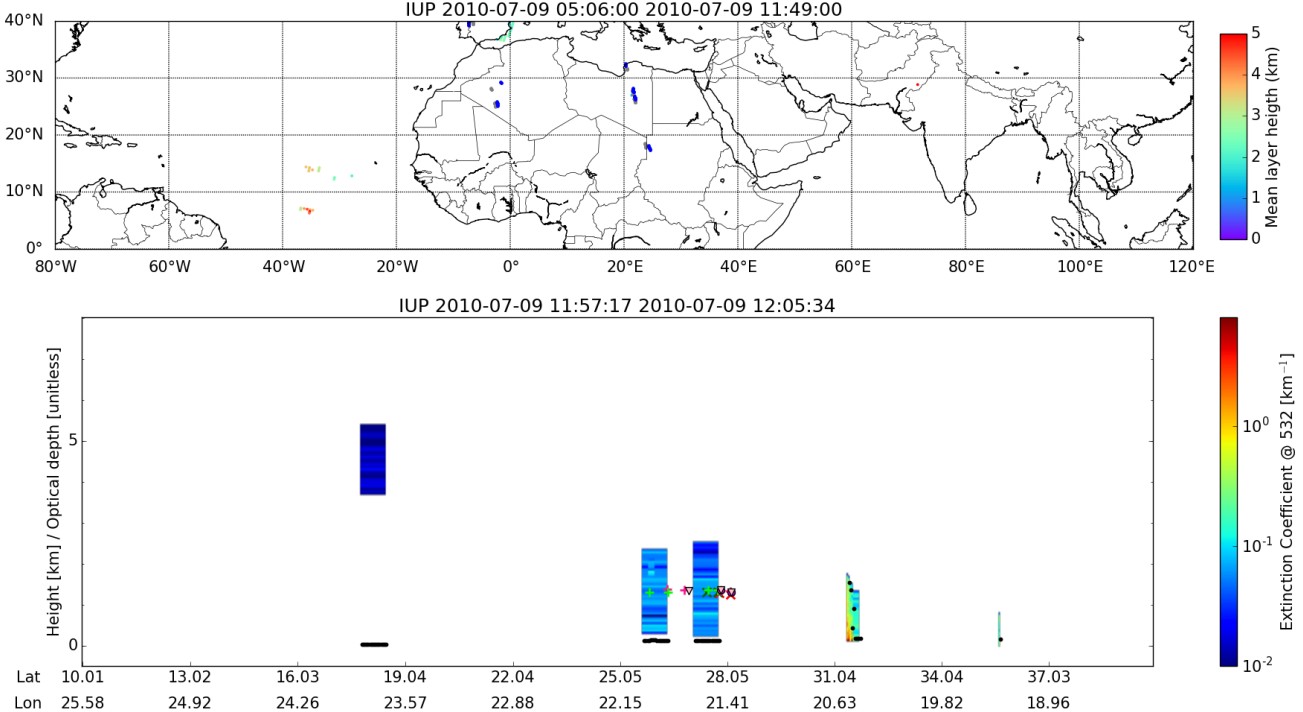

**Figure 7.** Similar to Fig. 2, but SCIAMACHY dust height from IUP analysis. Note difference in time from Fig. 2. The curtain in the bottom plot is for the CALIOP data between 20-40°E in top plot.

increases to -0.607 km when comparing with the CALIOP geometric mean height. For the LISA algorithm the behaviour is similar to the BIRA-IASB and DLR algorithms with mean height differences of -0.045 km (geometric mean) and 0.507 km (cumulative extinction). Scatter plots in Figs. 8-9 (upper panels) reveal rather elongated clouds of points along (parallel to) the 1:1 straight line for BIRA-IASB and LISA with respect to geometric mean (cumulative extinction) heights from CALIOP,

5 whereas the point cloud is mainly localized below 2 km of altitude for LMD and above 2.5 km for DLR (this last one presenting maxima of occurrences). The standard deviations are similar for the BIRA-IASB, DLR, LMD and LISA algorithms, between 1.029-1.187 km (geometric mean) and 1.126-1.339 km (cumulative extinction), being slightly lower for LISA, intermediate for BIRA-IASB and DLR, and to some extent greater for LMD.

Due to the larger footprint size for the solar sensors, fewer data points are available for dust height comparison of CALIOP

10 with GOME-2 and SCIAMACHY. Data for a single day are given in Figs. 6 (KNMI) and 7 (IUP). Note that the data are from different days. The statistics for all colocated GOME-2 and SCIAMACHY with CALIOP points are summarized in Fig. 10 and Table 3. Both algorithms give lower dust heights compared to CALIOP, with IUP being on the average lower by -1.097 km (-0.961 km) compared to the CALIOP geometric mean (cumulative extinction) height and KNMI lower by -1.393 km (-0.818 km).




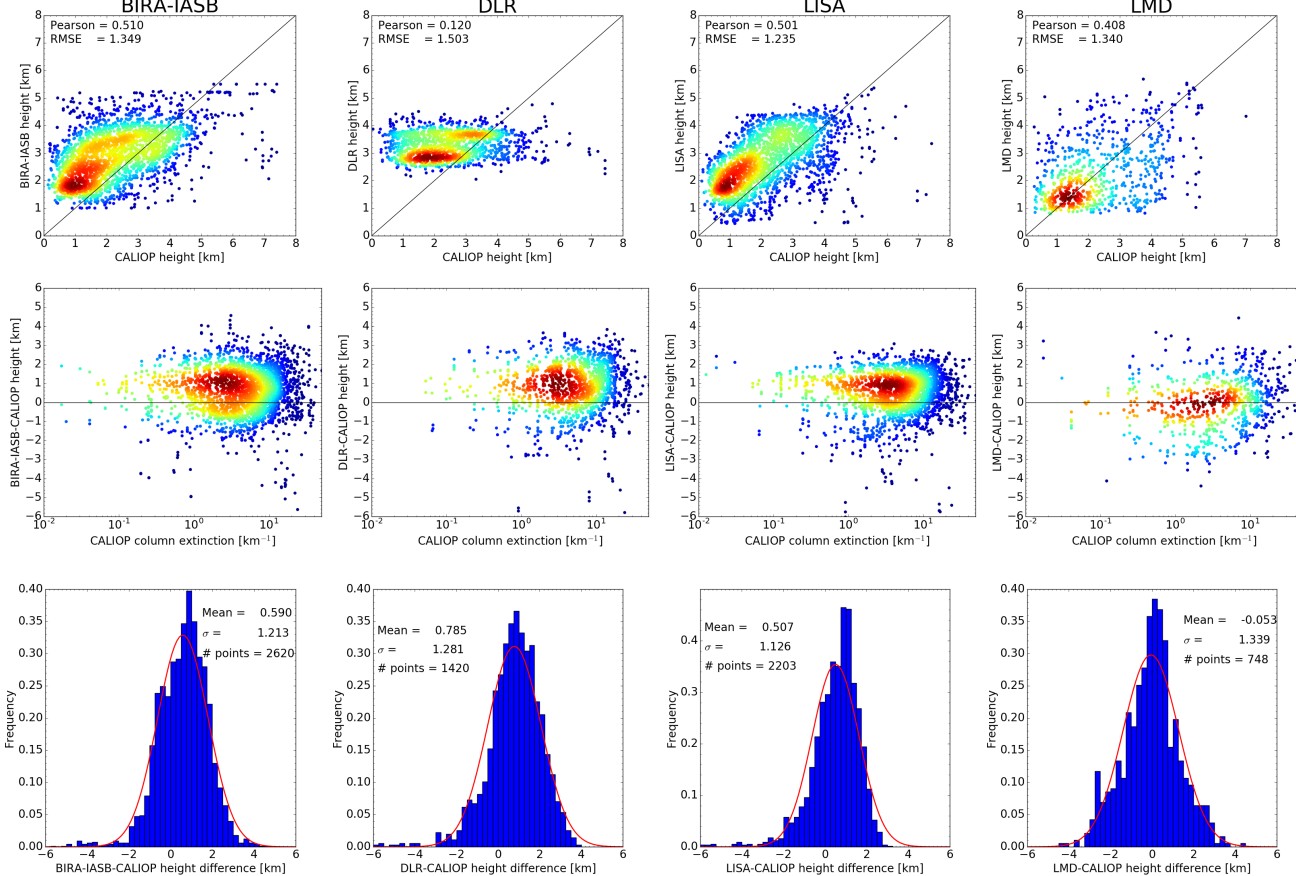

**Figure 8.** (Upper row) Scatter plots of the CALIOP cumulative extinction height versus height from the various algorithms. The color indicates the density of points. Also given are the Pearson's correlation coefficient and root mean square error (RMSE). (Centre row) The difference between passive algorithm and CALIOP cumulative extinction heights versus the CALIOP column extinction. The color indicates the density of points. (Bottom row) Frequency distribution of the difference between the height from the various algorithms and the CALIOP cumulative extinction height. Included is also a normal distribution fit to the difference. The mean and standard deviation ($\sigma$) of the normal distribution together with the number data points are given in each plot. This information is also provided in Table 3. Data are shown for the BIRA-IASB (first column), DLR (second column), LISA (third column) and LMD (fourth column) algorithms.

The features seen in the upper rows of Figs. 8-10 reveal that height differences may depend on region and time of day or other variables. It is well known that CALIOP daytime measurements are more noisy due to straylight from the sun. Thus to investigate possible differences between night and day time data the differences between CALIOP heights and the passive algorithm heights were calculated separately for night and day and also for land and ocean. For each IASI-algorithm plots

5   similar to those in Figs. 8-10 for ocean-day, ocean-night, land-day and land-night data subsets are shown in Figs. A1-A4 for



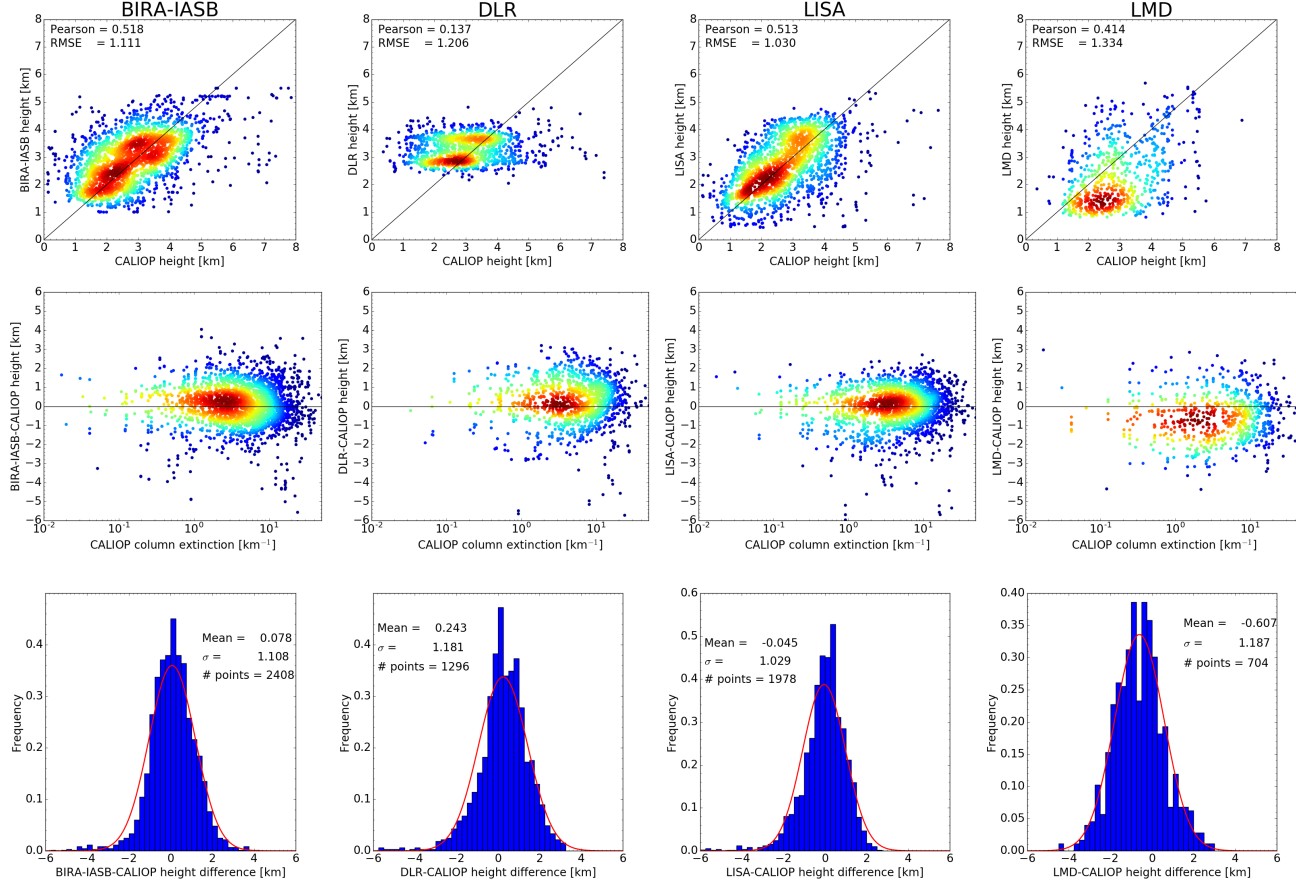

**Figure 9.** Similar to Fig. 8, but for the CALIOP geometric mean height This information is also provided in Table 3, numbers within parentheses.

the CALIOP cumulative extinction heights and in Figs. A5-A8 for the CALIOP geometric mean heights. The results are also summarized in Table 3.

For BIRA-IASB the mean difference is similar over land during day (0.087 km) and night (0.038 km) when comparing with the CALIOP geometric mean height. For the cumulative extinction height the mean difference increases from 0.357 km during day to 0.567 km during night over land. Over the ocean the mean difference is somewhat larger during the day (0.340 km) than night (0.094 km) for the geometric mean height, while it is vice versa for the cumulative extinction height, being 0.783 km (day) and 1.008 km (night). For DLR few data points are available over the ocean. Over land the mean difference is smaller for the day data than the night data, being -0.044 km (0.405 km) and 0.358 km (0.906 km) respectively for the geometric mean (cumulative extinction) height. For LMD the dust heights over land are found to be smaller than the CALIOP geometric mean (cumulative extinction) height during day than night, -0.496 km (-0.102 km) versus -0.579 km (0.073 km). Over the ocean the







**Figure 10.** Similar to Figs. 8-9, but for the KNMI (first and third column) and IUP (second and fourth column) algorithms versus the CALIOP cumulative extinction (first and second column) and geometric mean (third and fourth column) dust heights.

behaviour is similar, but the differences somewhat larger, see Table 3. The magnitude of the mean LISA difference is larger during the day (-0.635 km) than night (0.170 km) over land for the geometric mean height. For the cumulative extinction height the behaviour is opposite, being -0.225 km during day and 0.663 km during the night. Over the ocean a similar behaviour is observed. For nearly all comparisons the RMSE is smaller for the night data than the day data, most likely reflecting the lower

5   noise in the CALIOP night data.

The plots in Figs. A1-A8 reflect the findings presented in Table 3. The BIRA-IASB algorithm agrees well with the CALIOP geometric mean height over land for day and night, Fig. A1. Over ocean the agreement is better during the night. It is noted that for the ocean day subset the histogram is bimodal. When compared with the cumulative extinction height, Fig. A5, the BIRA-IASB dust height is overestimated over ocean during both day and night; the ocean day subset appears to be bimodal;

10   the agreement appears better over land during day than night, but this may in part be due to a bimodal histogram for the land





day subset. This is reflected in the spread in the difference which is smaller during night than day. For DLR, Figs. A2 and A6, there are few data points available over the ocean. Over land the DLR height data are clumped at a single height for the day subset and at two heights for the night data subset. For LMD, the agreement is monomodal for the land day, land night and ocean night subsets when compared with both the CALIOP cumulative extinction and geometric mean heights. For the

ocean day subset a bimodal distribution may be present. Overall the agreement is better when compared with the cumulative extinction height. The LISA data, Figs. A3 and A7, also have a bimodal ocean day distribution compared with the CALIOP heights. For the ocean the mean difference with the CALIOP cumulative extinction height is significantly larger during night than day. This difference is nearly a factor 2 smaller when comparing with the geometric mean height. For land the magnitude of the difference is smallest when compared with the cumulative extinction height during the day and with the geometric mean

height during night. The KNMI-GOME-2 dust heights compares better with the CALIOP cumulative extinction (geometric mean) dust heights over land, difference of -0.229 km (-0.893 km), than over ocean, -1.477 km (-2.015 km) Table 3.

The four dust episodes investigated may have dust with different optical characteristics that may have an effect on the retrieved dust heights. The comparison was therefore further subdivided into four time periods representing the episodes. In Figs. 11-12 is shown box, whisker and flier plots of the difference between the height from the various algorithms and the

CALIOP cumulative extinction height for the four episodes. There appears to be no clear temporal variations with medians within ±one quartile. The same conclusions are drawn when comparing to CALIOP geometric mean heights.

For the full data set mean height differences vary between -0.607 and 0.243 km (geometric mean) and -0.053 and 0.785 km (cumulative extinction) for the IASI algorithms and -1.393 and -1.097 km (geometric mean) and -0.961 and -0.818 km (cumulative extinction) for the solar algorithms, Table 3. In apricot color boxes in Table 3 are given the percentage of retrieved

heights from the passive sensors that are within the dust layer as seen by CALIOP. Here the CALIOP dust layer is identified as the lowermost and uppermost heights identified as dust. The highest percentage for the IASI algorithms is achieved by LMD with up to 68.8 % (geometric mean) and 48.5 % (cumulative extinction) dust heights located within the CALIOP dust layer during night over the land (for a subset of respectively 206 and 177 points).

The average CALIOP dust layer thickness is 0.925 km over land during day and 0.859 km over the ocean. For the night the

layer thickness is 1.292 km over land and 0.924 km over ocean. Thus, over land the dust layer thickness is larger during night than day by about 0.367 km while it is similar over the ocean. The dust layer over land is about 0.700-1.400 km higher over land than ocean and it is lower by about 0.700 km during the night than day. This is mainly caused by different regions being sampled during night and day overpasses. Most of the concurrent IASI and CALIOP night data are from the Persian Gulf and the Red Sea (lower dust height) while the day time data are more evenly distributed over the study area.

## 30  4  Discussion

We have compared dust layer heights from various passive sensors with CALIOP derived heights. The CALIOP heights are considered as the "true" values. However, the CALIOP heights are not unique as described in section 2.1.1, thus we have used two different CALIOP derived heights. For the cumulative extinction CALIOP height method, the lidar ratio is involved. This





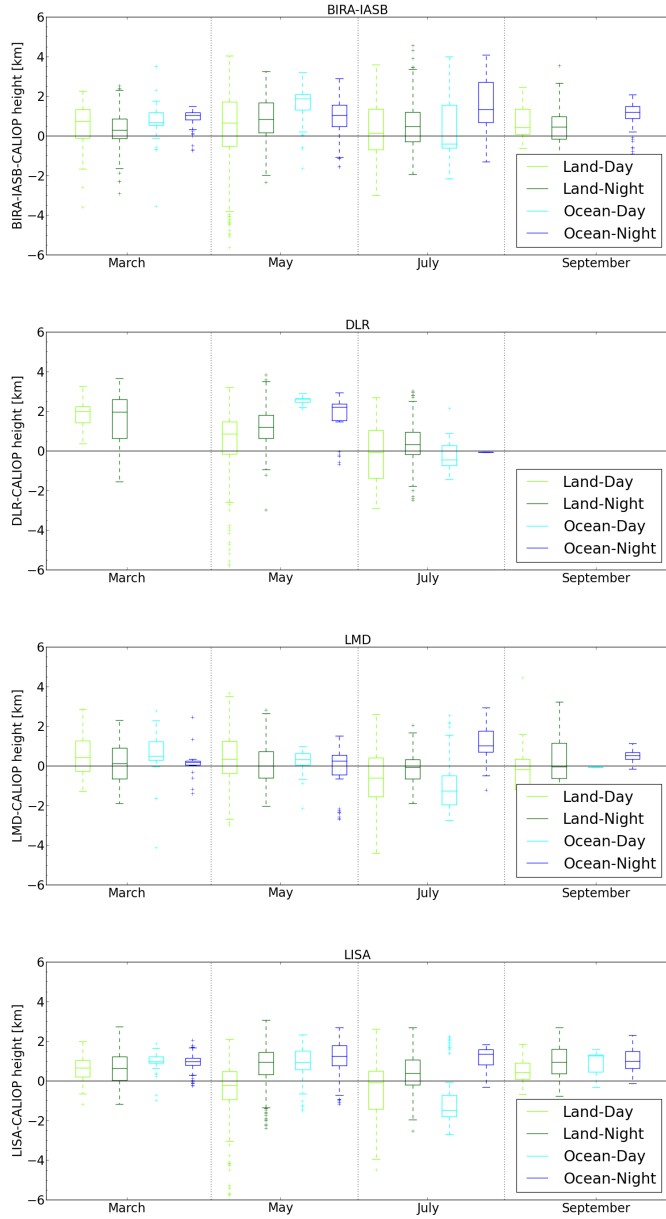

**Figure 11.** Box, whisker and flier plots of the difference between the height from the various IASI algorithms and the CALIOP cumulative extinction height. The box covers the lower ($Q_1$) to upper quartile ($Q_3$) values of the data. The horizontal line in the box is at the median. The whiskers show the range of the data where range is between $Q_1 - 1.5\Delta Q$ and $Q_3 + 1.5\Delta Q$, $\Delta Q = Q_3 - Q_1$. Flier points are those beyond the whiskers.





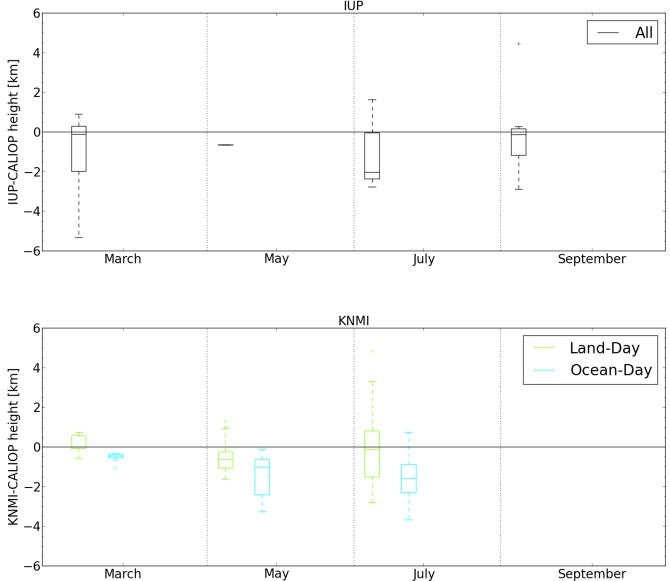

**Figure 12.** Similar to Fig 11, but for the IUP and KNMI algorithms.

may be different for different regions and time of year. Thus adding to the uncertainty in the comparison. The CALIOP analysis may also misclassify aerosol as discussed by Kim et al. (2013). The latter is largely avoided in this study by focusing on dust aerosol which have a relatively large depolarization ratio. Different methods to calculate CALIOP heights are compared in Fig. 1. The RMSEs for the height methods compared are 0.652 km and 0.182 km. These numbers should be kept in mind for the comparison results presented above.

While comparable, the heights retrieved from CALIOP and IASI are not the same quantities due to the instruments different sensitivities to various aerosol particle sizes and the assumptions of aerosol optical properties (lidar ratio, refractive index, particle shape) used in the retrieval. A full understanding of the reason for the differences requires a detailed algorithm comparison which is beyond the scope of this study. It is noted that infrared sensors have lower sensitivity to low height dust caused by the small temperature difference between the temperature of the surface and the temperature of the dust. For example for the BIRA-IASB algorithm the lowest possible retrieval height is around 1.2 km due to low sensitivity to dust at lower height. For the DLR algorithm a positive bias with respect to the CALIOP cumulative extinction height was predicted, section 2.2.2. A positive bias between 0.0405 (day) and 0.906 km (night) is indeed found over land surfaces, see Table 3.

Overall the standard deviation of the difference between CALIOP heights and the passive sensor heights, is smaller for the night time data than the day time data, Table 3. This is most likely due to less noise in the CALIOP night data. Standard deviations are generally similar for ocean and land data, but there are differences for individual algorithms indicating opportunities for future improvements.



There is quite a large difference between day and night over the ocean and all algorithms overestimate more at night than day over ocean, Table 3. Due to the differences in satellite overpass times different regions are sampled during night and day overpasses. Most of the concurrent IASI and CALIOP night data are from the Persian Gulf and the Red Sea (lower dust height) while the day time data are more evenly distributed over the study area. The differences seen between night and day data may thus be caused by differences in optical properties of the dust between the two regions, which is not accounted for by the retrieval algorithms.

The CALIOP heights are moved to the SCIAMACHY and MetOp-A overpass times. On average the vertical shift is small being between 0.015 and 0.020 km with a standard deviation of 0.25 km. For individual data points the shift may be larger, compare red and green crosses and lime and deep pink plusses in bottom panels of Figs. 2-7. This suggests that when comparing data sets from satellite sensors with different overpass times, transport processes should be accounted for in the analysis.

The spatial resolutions of the IASI, GOME-2 and SCIAMACHY instruments are much coarser than CALIOP. The impact of differences in spatial reolution has not been investigated, but it is assumed to be small within large dust clouds as studied here.

It is not straigthforward to estimate an uncertainty for the IASI height retrievals as this would require a sensitivity study that is beyond the scope of this work. A best-guess estimate would be that the uncertainty is on the order of 1-1.5 km.

Vandenbussche et al. (2013) found that for low dust loads, the BIRA-IASB algorithm placed the aerosol layer 1-2 km above the CALIOP retrieved layer. The algorithm has since then undergone several revisions and improvements and the average overestimate for all data is 0.078 km (0.590 km) when compared with the CALIOP geometric mean (cumulative extinction) height. Possible reasons for this overestimate is discussed above. Vandenbussche et al. (2013) reported better agreement for moderate to higher dust loads compared to low dust loads. In the present study no effect of dust load on dust height agreement appears to be present, see centre row plots of Figs. A1 and A5.

For monthly mean $1° \times 1°$ gridded IASI data covering the period July 2007-June 2013 Capelle et al. (2014) reported a systematic IASI-CALIOP bias of 0.4 km with a standard deviation of 0.48 km over the ocean. Peyridieu et al. (2013) reported similar values for the same data set. In this study for LMD over ocean a bias of -0.922 km (-0.501 km) against the CALIOP geometric mean (cumulative extinction) height is found for data recorded during the day overpasses with a standard deviation of 1.142 km (1.409 km). For the night overpasses the differences are -0.674 km (0.352 km) and the standard deviation 0.878 (1.180), Table 3. One reason for the larger spread in this study may be due to the use of monthly and spatially averaged data by Peyridieu et al. (2013) and Capelle et al. (2014) while here the comparison is made on a pixel by pixel basis. Hence extreme values are not averaged out.

Overall, for the IASI algorithms, two algorithms (BIRA-IASB and LISA) agree better with the CALIOP geometric mean height while LMD agree better with the CALIOP cumulative extinction height, Table 3. The DLR algorithm generally gives altitudes at two distinct modes, but overall agrees better with the geometric mean height. This may indicate that the IASI algorithms do not provide the same height information. The BIRA-IASB and LISA algorithms retrieve an aerosol profile from which a dust height is calculated. The LMD algorithm, on the other hand, use a single layer aerosol in the retrieval while DLR estimate the altitude from the retrieved dust layer temperature. The comparison with the two CALIOP heights suggests that the





profile retrieval is generally more sensitive to the actual dust layer vertical location. Both the BIRA-IASB and LISA algorithms use 1 km vertical steps. But with 1.5-2 degrees of freedom there is a significant correlation between the layers, and therefore a low sensitivity to the actual high resolution vertical distribution represented in the cumulative extinction height. Mean altitudes from those retrievals would then be something resembling a "geometric mean height". Contrary, the LMD algorithm, which

places the aerosol in a single homogeneous layer, is more sensitive to the aerosol layer radiatively "effective" height. It is also important to note that the sensitivity of the IASI algorithms does not only depend on aerosol load, but also on the temperature profile.

The BIRA-IASB algorithm use CALIOP profiles as a priori which implies that the BIRA-IASB altitude data include information about the CALIOP data against which it is compared. However, the (monthly) a priori is averaged over a large spatial

area ($5° \times 5°$) therefore including measurements from different days and most probably even different dust events. Furthermore, the retrievals usually have a significant departure with respect to the a priori. Thus the a priori profile used for a single retrieval is only vaguely related to the exact CALIOP profile used for the validation. The LISA algorithm also uses an a priori profile of dust derived from a CALIOP climatology, but it is a unique a priori profile for all retrievals. Therefore, it is not related with the CALIOP measurements used in the validation.

It is noted that all IASI algorithms assume the dust particles to be spherical. Klüser et al. (2016) compared optical properties of spherical and non-spherical dust particles. They found the values of the dust single scattering albedo to be different for spherical and non-spherical dust particles. This may potentially affect the dust height retrieval. It is beyond this study to investigate and quantify this effect.

The heights from the passive solar IUP-SCIAMACHY and KNMI-GOME-2 algorithms are generally low compared with

the CALIOP height, Table 3 . While IASI is mainly sensitive to the aerosol coarse mode, SCIAMACHY and GOME-2 are sensitive to both the fine and coarse modes. The height retrieved from these sensors depends on whether the surface albedo is retrieved simultaneously or not as shown by Sanders et al. (2015). They found that fixing the albedo in the retrieval gave a lower dust height than when retrieving both the albedo and the dust height. Fixing the albedo also gave better agreement with lidar measurements for the 16 scenes they analysed. We found the IUP-SCIAMACHY and KNMI-GOME-2 algorithm retrieved

heights to be low by -1.097 km (-0.961) and -1.393 km (-0.818) respectively when compared with CALIOP geometric mean (cumulative extinction) heights. For the KNMI-GOME-2 algorithm the underestimate is larger over ocean -2.015 km (-1.477) than over land -0.893 km (-0.229).

In general, possible reasons for the underestimation of layer height by the solar sensors are the local optical dust properties and the surface reflectivity assumed in the forward model. While it has been already demonstrated that a positive deviation

of the true surface albedo from the assumed prior value leads to an underestimation of layer height (Sanders et al., 2015), the similar tendency of lower retrievals by GOME-2 and SCIAMACHY suggest that the influence of a wrongly prescribed aerosol model can be ruled out. This is because the KNMI/GOME-2 algorithm uses a Henyey-Greenstein phase function whereas IUP-SCIAMACHY ingests spectrally resolved T-matrix calculations of the phase matrix representing aspherical dust particles (see Table 1).





To this end, we note that the algorithms of the solar spectrometers assume that satellite pixels are fully covered by dust. Because of their coarse footprint sizes, this condition can be frequently not satisfied. A situation of partially aerosol-covered pixels implies that less oxygen molecules are shielded by the intervening scatterers, with the effect of increasing absorption inside the A-band sensed by the instruments. This effect is even more pronounced closer to the ground, where the majority

of oxygen molecules reside. Since most of the information content on the height of the aerosol layer is carried by the in-band wavelengths of the A-band (about 760 nm) ratioed to the continuum (758 nm), it can be deduced that a dust pixel fraction smaller than one will lead also to an additional underestimation of layer height.

We end the discussion by listing several questions left open by this study. These questions may broadly be divided into two sets: 1) questions requiring analysis of larger dataset to consolidate findings; and 2) questions requiring a more detailed

analysis to better understand reasons for differences. Specific open questions are:

1A: How will the results change when including other types of aerosol in the analysis?

1B: How will a larger data set in time and space affect the results?

1C: May an optimal aerosol height algorithm covering all situations be developed?

2A: What are the physical reasons for the differences between the IASI-algorithms?

2B: What are the physical reasons for the differences between the solar algorithms?

2C: What are the physical reasons for differences between the quantities estimated by the IR and solar algorithms?

2D: How may synthetic data be used to understand and evaluate the various algorithms?

2E: Which pixel-level uncertainties can we estimate to the layer height results of each algorithm (based on studies 2A-2D)?

## 5   Conclusions

As part of the ESA Aerosol_cci project dust aerosol heights retrieved from passive infrared and solar sensors using different algorithms have been compared with two different CALIOP derived dust layer heights. The comparison was made on a pixel by pixel basis for the IASI, GOME-2 and SCIAMACHY sensors for four dust episodes in 2010. Time differences between the overpass of CALIOP and the passive sensors were accounted for by shifting the CALIOP heights to the location of the pixels of the passive sensors using the FLEXTRA trajectory model.

As it is not possible to construct a unique dust layer height from CALIOP data, two CALIOP derived layer heights were used: the cumulative extinction height which is set to the height where the CALIOP extinction column is half of the total extinction column; and the geometric mean height which is defined to be the geometrical mean of the top and bottom heights of the dust layer.

Four algorithms (BIRA-IASB, DLR, LMD, LISA) retrieved dust heights from IASI spectra. The mean difference between

the IASI heights and the CALIOP geometric mean (cumulative extinction) heights were found to vary between -0.635 and





0.087 km (-0.225 and 0.405 km) over land during day. For night time overpasses the values were -0.579-0.358 km (0.073-0.906 km). Over the ocean day differences were between -0.922-0.340 km (-0.501-0.913 km) and night time differences between -0.674-0.835 km (0.352-1.599 km). Standard deviations were between 1.322-1.572 km (1.448-1.665 km) over land during day and decreasing to 0.855-1.058 km (0.896-1.092 km) during night. Over the ocean the standard deviation decreased

from 1.142-1.187 km (0.913-1.539 km) during the day to 0.486-0.878 km (0.637-1.180 km) at night.

Two of the IASI algorithms (BIRA-IASB and LISA) were found to agree better with the CALIOP geometric mean height (BIRA-IASB: 0.078 km (cumulative extinction 0.590 km); LISA: -0.045 km (0.507 km)) while the LMD algorithm agreed better with the CALIOP cumulative extinction height: -0.053 km (geometric mean: -0.607 km). This is believed to be caused by the differences in the aerosol profile used for the radiative transfer simulations: BIRA-IASB and LISA use and retrieve

vertically extended and resolved profiles while LMD place all the aerosols in one single homogeneous layer.

Far fewer data points were available for the solar sensors due to their larger pixel size and lack of night time data. The heights retrieved from the solar sensors on the mean underestimate the CALIOP geometric mean (cumulative extinction) heights by -1.393 km (-0.818 km) (KNMI, GOME-2) and -1.097 km (-0.961 km) (IUP, SCIAMACHY). This may be caused by the large pixel size and the assumption in the retrieval that the pixels are fully covered by aerosol.

The IASI instrument was first flown in 2006 and was the first of several to be launched. Thus data from IASI has the potential to provide long global time series of ECVs. There is considerable variation between the IASI retrieved dust heights. Nevertheless, if careful consideration is taken of differences in temporal and spatial characteristics of the observations, it might be feasible to construct a global data set of IASI retrieved heights quality controlled against CALIOP. The quality control will allow uncertainties on a pixel by pixel basis which again may be used for sensitive studies. Such a dust height data set may

be used to further our understanding of dust on the climate system. However, several open questions should be answered to have a better understanding of the quantities measured and their accuracy. A list of open questions are given at the end of the discussion section and includes both studies requiring large data sets and time periods, and studies looking at algorithm specifics.

Finally, the various algorithms and instruments are different in their approaches to retrieve the dust height. In the comparison

with CALIOP no single algorithm is found to stand as the best overall. Different methodologies may give best results at different locations and situations. Thus it seems fruitful to continue development of all algorithms and encourage comparison exercises.

*Code and data availability.*   All data are available to registered users from http://www.icare.univ-lille1.fr/. The FLEXTRA model is available from https://www.flexpart.eu/.

*Competing interests.*   The authors declare that they have no conflict of interest.




*Acknowledgements.* This work was supported by the European Space Agency as part of the Aerosol_cci project (ESA Contract No. 4000109874/14/I-NB).





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

## Appendix A: Additional figures

In Figs. A1-A4 statistics are shown for all and land-day, ocean-day, land-night and land-day subsets for all IASI dust height retrieval methods as compared with the CALIOP geometric mean heights. Figs. A5-A8 show similar data but using the CALIOP cumulative extinction heights.





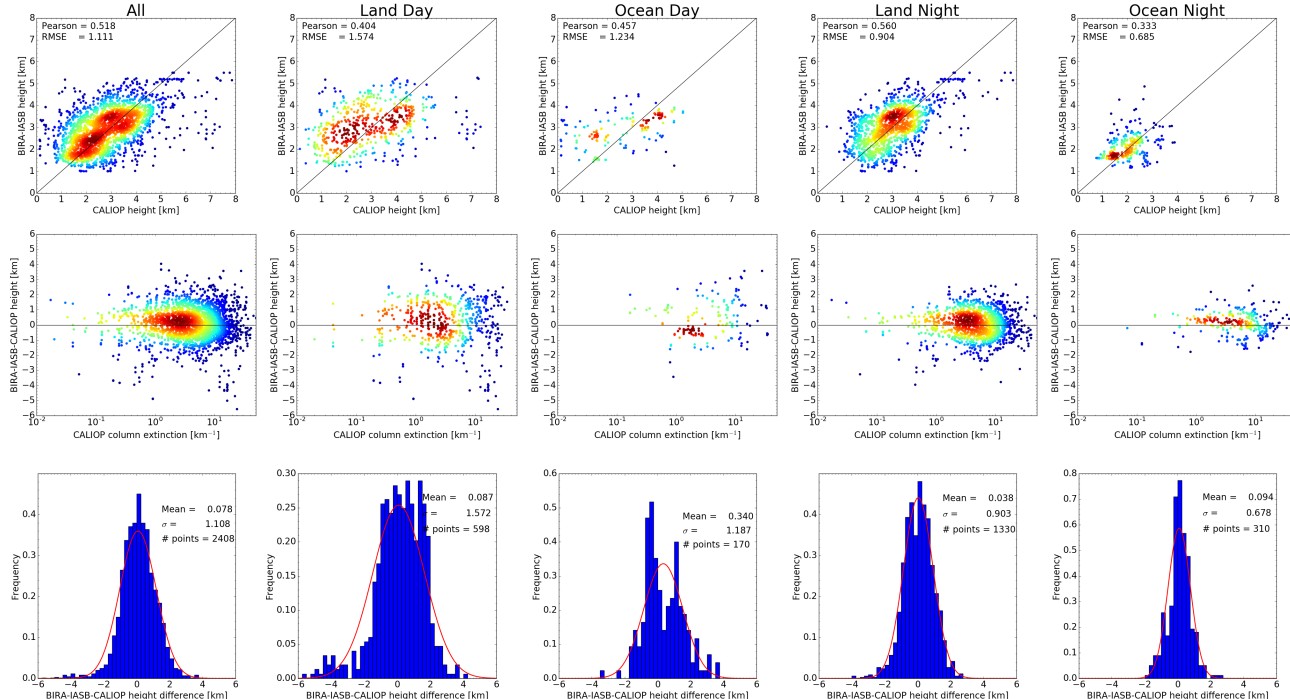

**Figure A1.** (Upper row) Scatter plots of the CALIOP geometric mean height versus height from the BIRA-IASB algorithm. (Centre row) Scatter plot of the difference between the BIRA-IASB and CALIOP heights versus the CALIOP column extinction. (Bottom row) Frequency distribution of the difference between the height from the BIRA-IASB algorithm and the CALIOP height. Included is also a normal distribution fit to the difference. The mean and standard deviation of the normal distribution are given in each plot. (Column 1) All data points. (Column 2) CALIOP day data and IASI land pixels. (Column 3) CALIOP day data and IASI ocean pixels. (Column 4) CALIOP night data and IASI land pixels. (Column 5) CALIOP night data and IASI ocean pixels.





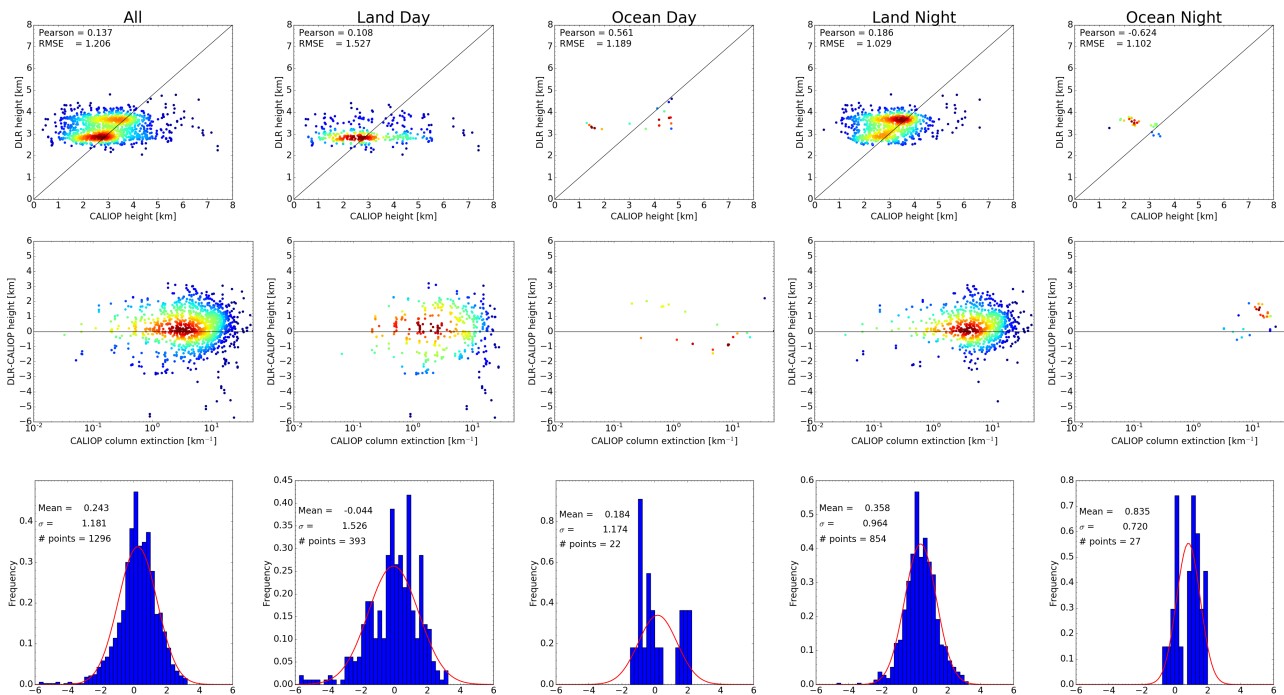

**Figure A2.** Similar to Fig. A5 but for the DLR algorithm.




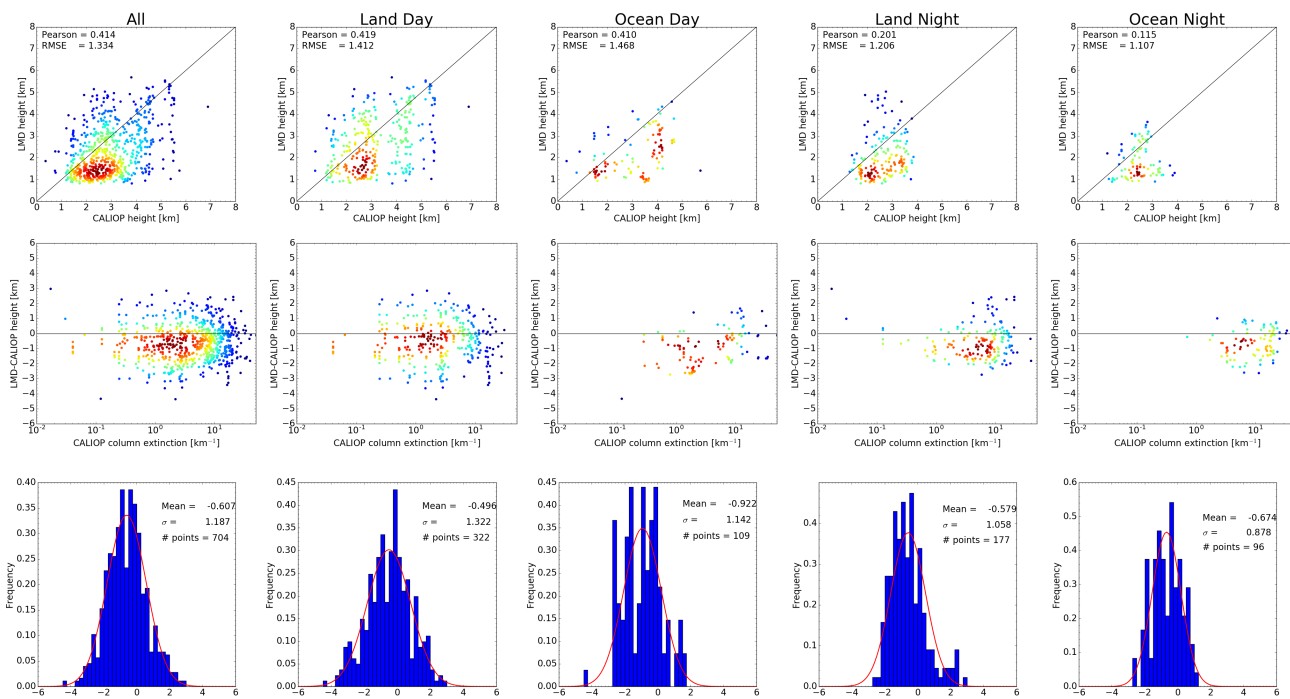

**Figure A3.** Similar to Fig. A5 but for the LMD algorithm.




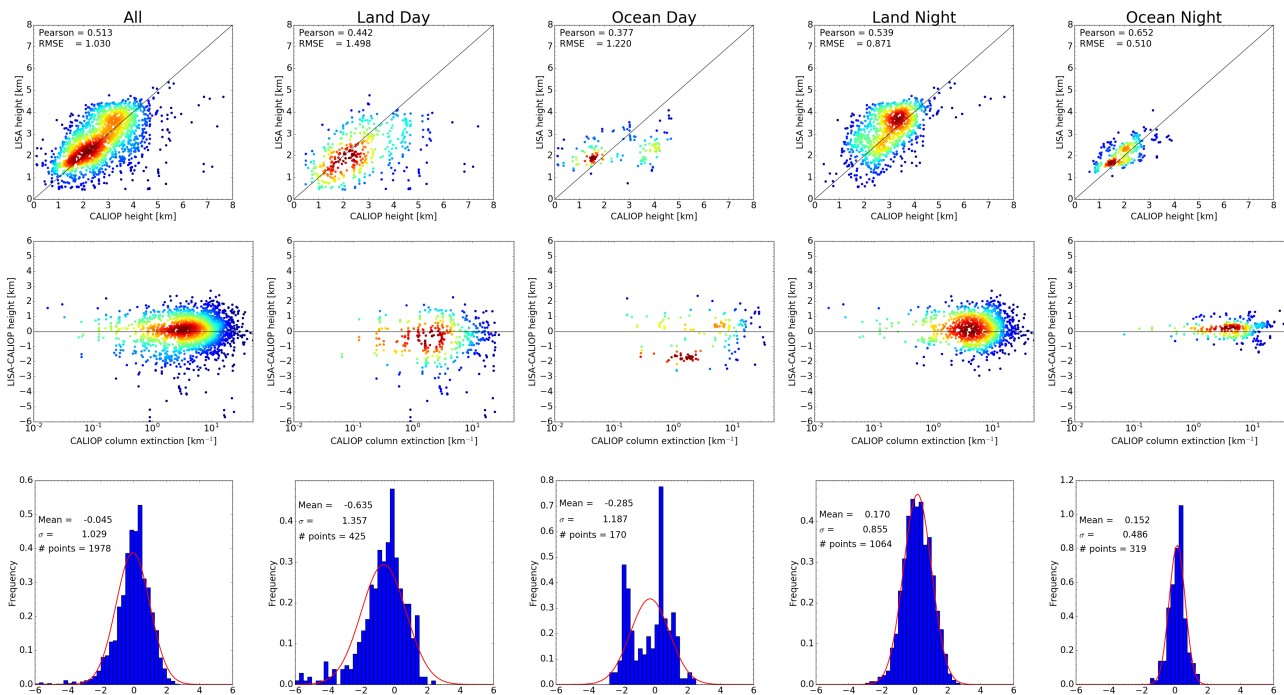

**Figure A4.** Similar to Fig. A5 but for the LISA algorithm.





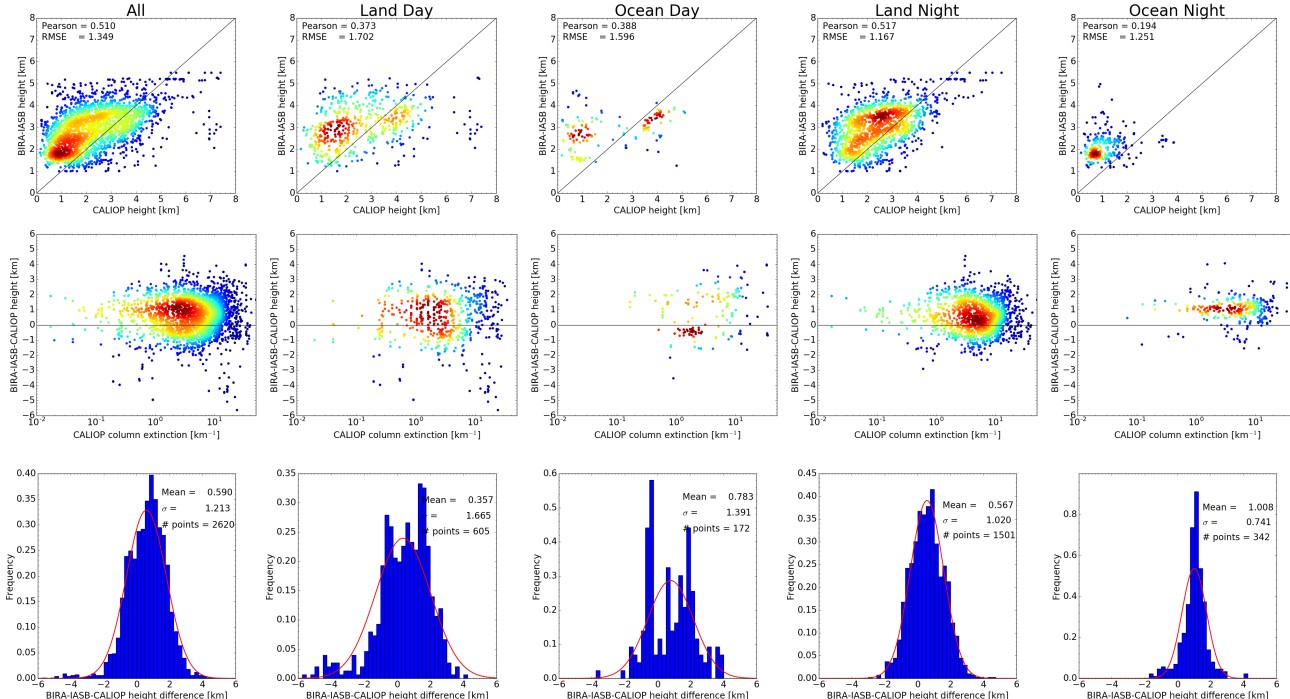

**Figure A5.** (Upper row) Scatter plots of the CALIOP cumulative extinction height versus height from the BIRA-IASB algorithm. (Centre row) Scatter plot of the difference between the BIRA-IASB and CALIOP heights versus the CALIOP column extinction. (Bottom row) Frequency distribution of the difference between the height from the BIRA-IASB algorithm and the CALIOP height. Included is also a normal distribution fit to the difference. The mean and standard deviation of the normal distribution are given in each plot. (Column 1) All data points. (Column 2) CALIOP day data and IASI land pixels. (Column 3) CALIOP day data and IASI ocean pixels. (Column 4) CALIOP night data and IASI land pixels. (Column 5) CALIOP night data and IASI ocean pixels.




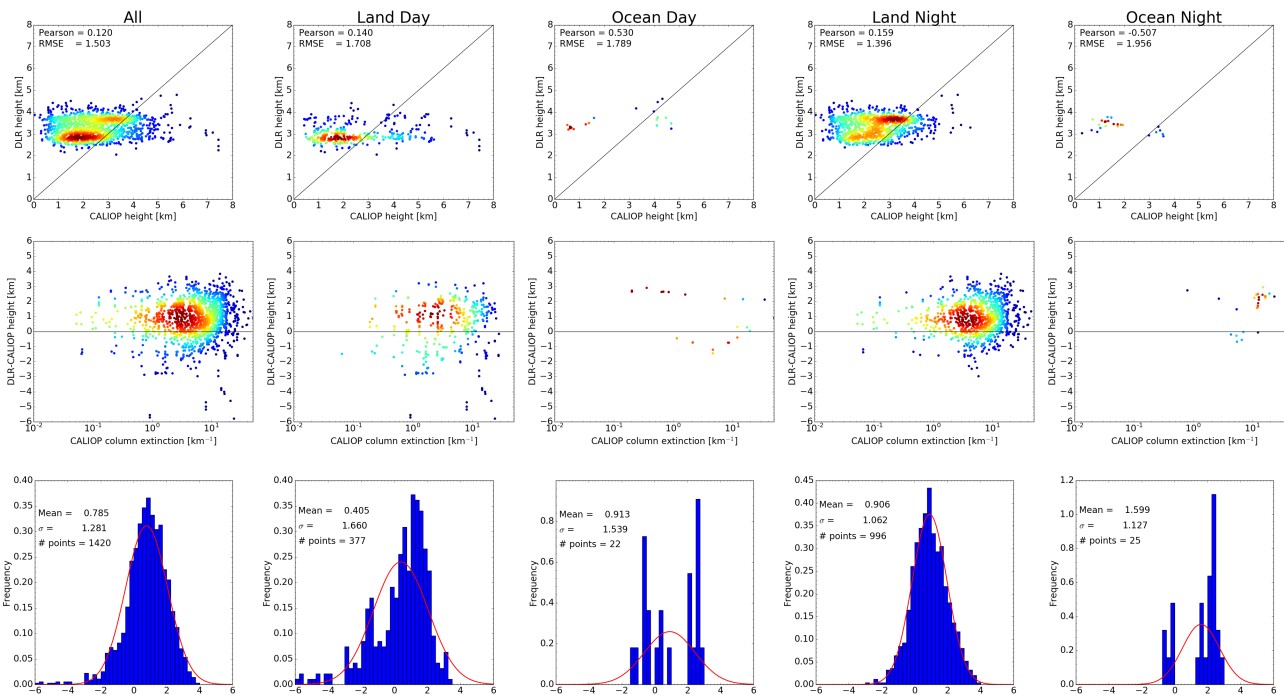

**Figure A6.** Similar to Fig. A5 but for the DLR algorithm.




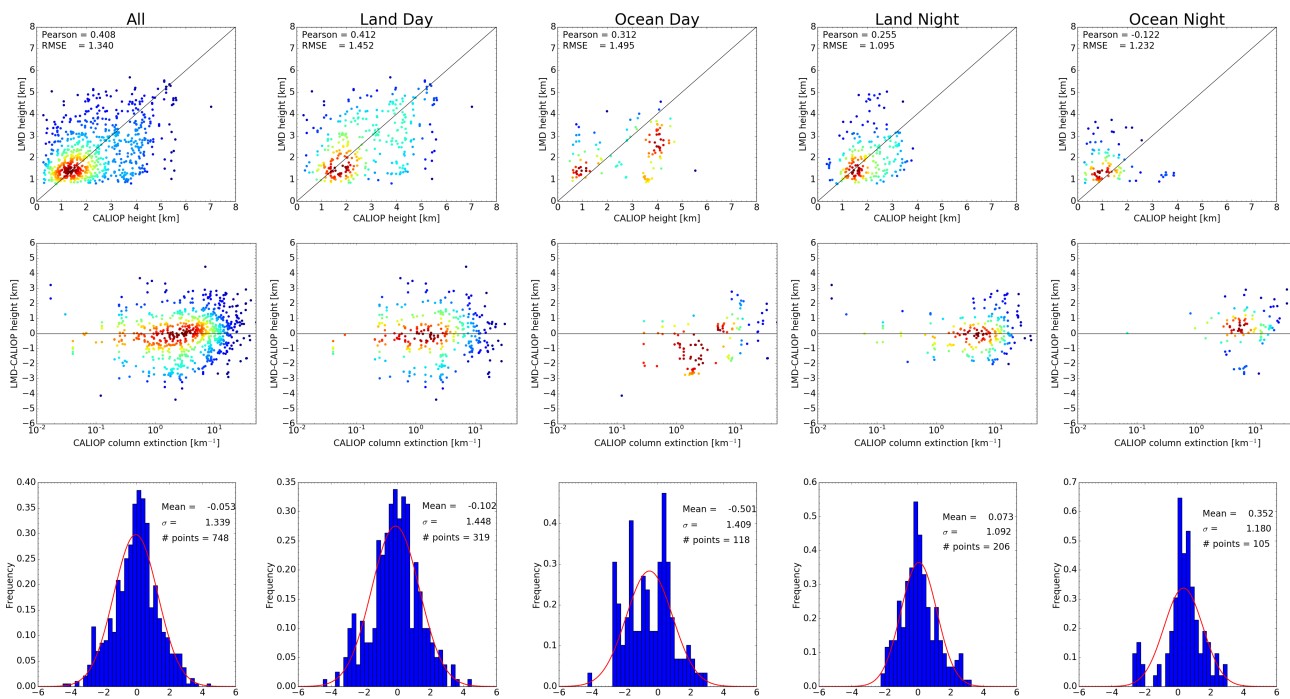

**Figure A7.** Similar to Fig. A5 but for the LMD algorithm.





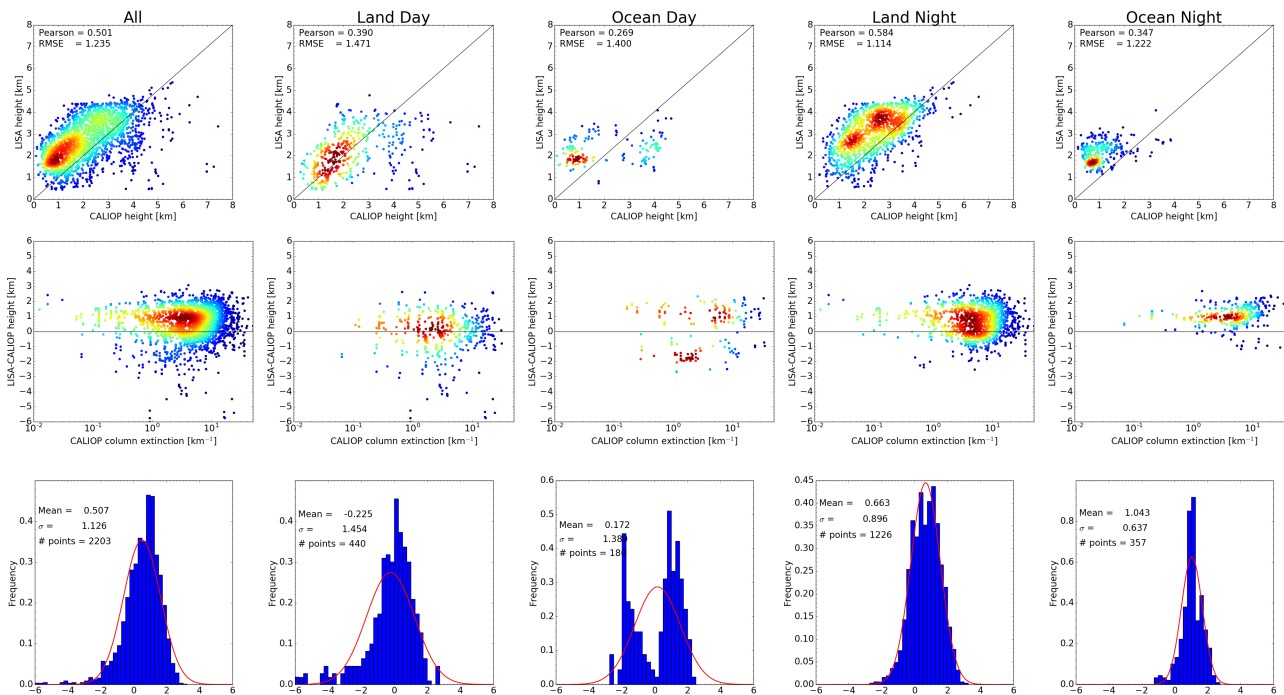

**Figure A8.** Similar to Fig. A5 but for the LISA algorithm.