# Peer review of "Comparison of dust layer heights from active and passive satellite sensors"

_Atmospheric Measurement Techniques, 2017_

## Referee Comment (RC1) · Anonymous Referee #1 · 2 Nov 2017

This paper compares dust aerosol layer height from passive sensors (different IASI algorithms, as well as GOME-2 and SCIAMACHY) against active observations from the CALIPSO lidar. This work is important and relevant to the scope of AMT, as aerosol height affects factors like radiative effects, transport, and possible contribution to surface air quality, yet aerosol height is less readily available from satellite sensors compared to quantities like AOD. Previous work mostly considered gridded data or more limited case studies, so this expands the volume and resolution of data considered. As the thermal sensor (IASI) is only sensitive to coarse aerosols, the analysis is restricted to dust layers. I like that the authors examine different reasonable definitions of 'aerosol layer height' from CALIPSO (half cumulative extinction vs. geometric mean), and that they attempt to account for movement caused by time differences between

[Figure]

CALIPSO (early afternoon orbit) and the other sensors (mid-morning orbits). I also like Table 1, which provides a direct comparison of various key features of the algorithm and references. This made it easy to see how the algorithms are similar and how they are different. The conclusions seem to be that the typical level of error in these heights is of order 1-2 km, but exact errors and biases depend on how the CALIPSO height is defined, underlying surface, and (for IASI) whether the scene is daytime or nighttime.

I don't have any major scientific problems with this paper (aside from more information about FLEXTRA being needed and an omission of discussion of uncertainty estimates; see later). I think it hangs together quite well, and leaves open questions for future study in the Conclusions. I recommend re-review after major revisions, in large part to improve the presentation of the paper (Figures and Tables are hard to follow and need reorganising), because I feel I need to see the redrawn Figures and Tables to make sure that my interpretation of what is shown and discussed is correct. My specific comments are below (PX LY refers to page X, line Y of the submitted manuscript):

Abstract: the quantitative discussion here only really covers the IASI data sets. A sentence or two about GOME-2 and SCIAMACHY performance could be added as well.

Table 1: I suggest changing the 'institute' column to cover both the institute and the algorithm name, as it is algorithm names (e.g. MAPIR, IMARS) which are often referred to in other publications and may be most memorable.

P5L9: the MAPIR data presented are stated to be from two different versions, v3.2 and 3.4. However, it is hard to tell how significant the differences are. From the text it seems to only affect the a priori temperatures used in the retrieval, but it is not clear if this is the only difference. This should be clarified in the text. If the difference in retrievals from the change to the prior is negligible, then this should be stated and the sentence reworded to avoid potential reader confusion/concern. My suggestion, both for simplicity and consistency (both internal and with any future available data set)

would be to just reprocess all the data with the latest version of the algorithm (v3.4).

P7, general: some of this information can be removed for readability, since there is already a reference for the algorithm in table 1 and some of this information is probably not directly necessary for the interpretation of the results here. For example I don't think the section from L7-L18 about the a priori can be shortened or removed, unless the authors feel this is significantly different from what is described in the prior algorithm paper or is somehow crucial for the understanding.

Section 2.2.4: this algorithm description can also be shortened, as Table 1 provides a reference and some details for the interested reader, and much of the information provided is not necessary for the interpretation of the results. For example P10L2-7, P10L18-22.

P13L10-24: from this it seems like the movement did not make a large systematic change to the dust height (0.02 km or less). Since the standard deviation of the height change was 0.25 km, this also implies that the error which would result from not attempting to account for the time difference is small, and that the altitude of dust layers is fairly stable over the time period of (as I understand) up to 5 hours. So that in itself is an interesting result as it suggests temporal sampling errors are not a big problem for this type of comparison, at least when looking at the big picture (since 0.25 km is somewhat smaller than the other retrieval uncertainties). The other side of the coin, which I think should be discussed more, is whether FLEXTRA's assessment of the transport during this time period is accurate. I would like to see some quantification on the reliability of FLEXTRA here as I assume it depends in part on the resolution and quality of the meteorology data ingested, since this is what will determine the horizontal and vertical motion from the trajectory. So for example which ECMWF data version was used? Has FLEXTRA been quantitatively evaluated? If you use another meteorological data set or change some other input parameter is the solution stable? I know that e.g. HYSPLIT lets you run an ensemble of trajectories as one way to assess stability; I don't know if FLEXTRA does this too. The bottom line is that the comparability between

CALIPSO and other sensors rests on the use of FLEXTRA to assess and correct for the change in aerosol location between observations, and this is not covered in much detail in the present version of the manuscript, so more discussion is needed.

Figures 2-7: I understand the intention here, but find these figures hard to interpret. In the top panel the grey is hard to see unless you zoom a lot, and the blue lines are somewhat similar to tones in the colour table. For the bottom panels, the background lidar curtain is the same between the IASI figures but there are too many different coloured symbols overlaid to quickly and easily compare and get the picture of what is going on. I wonder if, for the IASI plots at least, these could be redrawn. For example, make the top panels of Figures 2-5 their own figure, since these are all the same date and same map bounds, so we can directly compare the maps of coverage and heights between the different algorithms for IASI. Then also make the bottom panels their own figure and simplify in some way for clarity. For example we probably don't need to see the lidar curtain (although this could be added as a panel by itself at the top, or as a separate figure) since what is being compared is the effective heights, and the curtain just adds 'noise' when the eye is trying to see coloured symbols. The deep pink pluses and red crosses appear similar in tone, again taking the lidar curtain off would allow one to use a more contrasting color for one such as blue or black. The points for CALIPSO AOD could also be removed since this is in black and naturally the most eye-catching, yet it is the least relevant since it is one set of points which does not represent a height. (In my mind the think you want the reader to focus on should probably be black or red since these stick out.) Alternatively the CALIPSO AOD could be just overlaid on the lidar curtain. The point here is that when looking at figures like this I want to answer two questions: (1) how similar are the spatial coverage and distribution of heights from the data sets and (2) how do they compare to CALIPSO? The present Figures do not allow me to do this effectively.

I realise that Figures 6 and 7 are a different date so can't be combined with the IASI figures for my suggested redrawing. Is there no date with all instruments providing

data? If so that would be better to show. Since both have much sparser coverage than the IASI plots, though (both these Figures are essentially a lot of white space and then overlapping symbols which are hard to distinguish without zooming), unless a better common case can be found, perhaps these two examples could be removed. I understand that from parity you probably want to show one case from each algorithm, but surely there must be a more instructive case than this; there is so little data that it's hard to assess looking at it whether these data sets are reasonable or not. And the summary statistics are in Table 3, so if the data are always sparse I don't know that we need to see GOME/SCIAMACHY case studies.

Table 3: as with Table 1, I'd add algorithm name here in the column headers. I'd also add instrument name, for ease of reference. I also have some formatting questions/comments here. In general this table is not well organised because the values in the headers only seem to refer to some of the statistics (I guess the top four of the eight boxes in each subset?) and there are too many colours. I think this could be much simplified, and the clarity improved, by condensing things and labelling individual rows rather than relying on colour-coding and large column headers. If I interpret the legend right, the top left and top right numbers in each column are the mean and standard deviation of height difference. So this could be represented as one entry, e.g. 0.590 +/- 1.213 km for BIRA-IASB. And the row legend would just read "height difference, km". Then you'd have other rows for "number of points" and "inlay". Labelling rows (which would then repeat for the all points, day/night, land/ocean splits) would remove the need for the complicated colour coding, and the header would only say instrument and institute/algorithm name. The statistics for cumulative vs. geometric heights could be split as left-right subcolumns for each algorithm instead. Does this make sense? Changing to that layout would make each row/column's content unambiguous, and dispense with the need to have 10 different colours (plus white) to code the table. As it stands, again, it is very difficult to pick out the key numbers from the table.

Figure 8 (and later): if these are scatter density plots on the first two rows then shouldn't

they be shown as filled rectangles rather than clouds of points? There is also no colour scale to indicate what is shown (absolute or relative frequency, how many points are we looking at)? I suggest redrawing. For the bottom panels, it would be clearer if a vertical line for a height difference of 0 km is added. This will again aid in direct comparisons of the data sets. I would take off the Normal distribution fits; they don't add anything that we can't already see from the bins, and don't appear to be great fits in some cases anyway (the actual distributions seem to have higher kurtosis than a Normal distribution, at least for IASI).

Figures 11, 12: I don't think these add anything to the discussion, and should be deleted. This is essentially another visualisation of the data already presented in Figures 8-10 and Table 3, albeit also sliced by time. The lines are also too faint to see. The authors can just modify the discussion on P24, L12-16 to note that optical characteristics may have been different but no clear temporal variation was found when the data were examined. We don't need to see the plots.

P24L19-23: From the 'inlay' columns it seems overall that this has values between 5.3% (LISA, ocean, night, cumulative) and 68.9% (LMD, land, night, cumulative), with typical values being something of order 30%. That means that typically two thirds or so of retrieved heights are entirely outside the dust layer (i.e. the retrievals put the dust somewhere totally without dust). Is that interpretation really correct? If so, that sounds pretty bad, and seems at odds with the other statistics presented, which show a standard deviation of around 1 km or so (and to me seems like a good result). I would double-check the calculation of this "inlay" statistic or reword the text if I have misunderstood what it means. The only way I can think of to reconcile this discrepancy is if the vertical extent of the CALIPSO dust layers is typically significantly smaller than the roughly 1 km IASI retrieval error. So perhaps some statistics about the CALIPSO dust layer geometric thickness should be presented here. Either way, there appears to be some unresolved issue in this statistic or its interpretation.

P27L14-15: Table 1 indicates that several of the algorithms use the Optimal Estimation Method (OEM) or similar techniques, which should be able to provide pixel-level uncertainty estimates on retrieved aerosol height. It would be instructive to compare these estimates in a statistical sense to the level of agreement with CALIPSO, to see whether these uncertainties are reasonable. This does not appear to have been done; I suggest adding it in the revised manuscript, since this analysis provides a useful way to 'validate' the uncertainty estimates. I realise that this can't be done for all the data sets, but since this is a big advantage of OEM, it makes sense to use it! For the OEM retrievals, it should be there already. If it is not, why not? It is definitely naturally within the scope of the existing study.

---

## Referee Comment (RC2) · Anonymous Referee #3 · 27 Mar 2018

This paper compares dust layer heights retrieved from various passive remote sensing algorithm to those detected by an active space lidar, the CALIOP. In particular, authors considered four IR algorithms applied to IASI measurements and two oxygen-band algorithms applied to GOME-2 and SCIAMACHY measurements in the O2 A bands. While aerosol height is the one of the most important variables that determine how aerosol affect climate and air quality, passive remote sensing of aerosol height is extremely more challenging than sensing the aerosol loading. This paper thus address an important study and falls into the scope of AMT journal. I agree with the first reviewer that, while the science and used approaches are sound, this manuscript still needs to improve its presentation of the results and discussion. And, a major revision is necessary to improve the logic and clearness of the paper.

Major comments:

1. I have the same feeling to the first reviewer that Figure 2-7 and their discussions are difficult to follow. I also suggest Figures 2-5 and 6-7 be merged, so that readers can easily compare the spatial distribution of heights between different algorithms. I would also recommend a legend be added to the curtain plot to indicate the meanings of each symbol.

2. Table 3 is particularly hard to follow. I recommend, instead of using table, use bar plots to compare those statistics to different algorithms.

3. Too much text is used to present Figures in the Appendix. Those figures should be briefly mentioned, so only the major findings from them be presented.

4. This study found that solar algorithms yield larger bias (> 1 km) for the case of dust aerosol height than the IR algorithms. However, it should be noted that some studies have shown an accuracy of about 0.5 km of dust layer height from O2-A might (studies listed below). So authors may need to compare and justify the performance of the current study to those studies:

Kokhanovsky, A. A., and V. V. Rozanov (2010), The determination of dust cloud altitudes from a satellite using hyperspectral measurements in the gaseous absorption band, International Journal of Remote Sensing, 31(10), 2729-2744, doi:10.1080/01431160903085644.

Dubuisson, P., R. Frouin, D. Dessailly, L. Dufor $\sqrt{}^{TM}$ t, J.-F. B. L $\sqrt{}^{CO}$ on, K. Voss, and D. Antoine (2009), Estimating the altitude of aerosol plumes over the ocean from reflectance ratio measurements in the O2 A-band, Remote Sensing of Environment, 113(9), 1899-1911, doi:10.1016/j.rse.2009.04.018.

Xu, X., J. Wang, Y. Wang, J. Zeng, O. Torres, Y. Yang, A. Marshak, J. Reid, and S. Miller (2017), Passive remote sensing of altitude and optical depth of dust plumes using the oxygen A and B bands: First results from EPIC/DSCOVR at Lagrange-1

AMTD
point, Geophysical Research Letters, 44(14), 7544-7554, doi:10.1002/2017GL073939.

Specific comments:

P2, L11: I'd to bring an attention to a recent review article about passive remote sensing of aerosol height by Xu et al. 2018, which is worth to cite: Xiaoguang Xu, Jun Wang, Yi Wang and Alexander Kokhanovsky, Chapter 1 - Passive Remote Sensing of Aerosol Height, In Remote Sensing of Aerosols, Clouds, and Precipitation, Elsevier, 2018, Pages 1-22, ISBN 9780128104378, https://doi.org/10.1016/B978-0-12-810437-8.00001-3

P3, L5-10: It mentioned here that these selected dust events are of Saharan origin, but the studied area are also frequently affected by dust emitted from Middle East, India, and Western China. Please be accurate.

P14, Figure 2(bottom): Symbols are hard to follow. A legend may be added to indicate the meaning of each symbol.

P15, Table 2: I don't quite understand the bracketed numbers in the third and forth rows. Please clarify in the Table caption (or using table footnote, as the caption is already very long).

P21, Figure 8: A colorbar is needed for the density of the scatters (similarly in Figure 9-10). The definition of the density is also necessary in the figure caption.

---

## Author Response (AR1)

**Response to interactive comments from Referee #1**

The referee is thanked for the careful reading of and constructive comments to the manuscript. The referee's comments are repeated below in italic font. The responses to the comments are shown in roman font.

**Comments**

- *Abstract: the quantitative discussion here only really covers the IASI data sets. A sentence or two about GOME-2 and SCIAMACHY performance could be added as well.*

  We have added the following text to the Abstract to include GOME-2 and SCIA-MACHY performance. "For the solar sensors it is found that SCIAMACHY data on the average are lower by -1.097 km (-0.961 km) compared to the CALIOP geometric mean (cumulative extinction) height and GOME-2 lower by -1.393 km (-0.818 km)."

- *Table 1: I suggest changing the institute column to cover both the institute and the algorithm name, as it is algorithm names (e.g. MAPIR, IMARS) which are often referred to in other publications and may be most memorable.*

  The algorithm names have been included in tables 1, 2 and 3 as suggested.

- *P5L9: the MAPIR data presented are stated to be from two different versions, v3.2 and 3.4. However, it is hard to tell how significant the differences are. From the text it seems to only affect the a priori temperatures used in the retrieval, but it is not clear if this is the only difference. This should be clarified in the text. If the difference in retrievals from the change to the prior is negligible, then this should be stated and the sentence reworded to avoid potential reader confusion/concern. My suggestion, both for simplicity and consistency (both internal and with any future available data set) would be to just reprocess all the data with the latest version of the algorithm (v3.4).*

  Indeed, the only difference between the two versions is the a priori surface temperature. The reason for this change is that for periods in time when the IASI level 2 data from EUMETSAT is in version 4, the retrieved surface temperature is often completely off, especially over deserts, therefore being a bad a priori for our retrievals. In those cases, the ECMWF ERA-interim reanalysis makes a better a priori. For later versions of the EUMETSAT IASI level 2 product, the retrieved surface temperature makes a better a priori for MAPIR than ERA-interim (closer to truth, and also easier to use). The change between EUMETSAT IASI level 2 version 4 to 5 occurred on 14 september 2010, so it concerns the 4th period in the current analysis. We have tested that using the a priori surface temperature from ECMWF ERA-interim or from EUMETSAT IASI level 2 version 5 is indeed negligible.

  As there is now a publication available that contains the full algorithm description, where we have used the MAPIR version name 3.5 to join together versions 3.2

and 3.4 for simplicity, we have changed this here too. The discussion of the Ts a priori has been removed from here, as it is not required for this manuscript, has little impact on the results, and may now be found in the other manuscript which reference is provided.

- *P7, general: some of this information can be removed for readability, since there is already a reference for the algorithm in table 1 and some of this information is probably not directly necessary for the interpretation of the results here. For example I dont think the section from L7-L18 about the a priori can be shortened or removed, unless the authors feel this is significantly different from what is described in the prior algorithm paper or is somehow crucial for the understanding.*

Indeed we agree that there was too much information in that section. It was due to the fact that the algorithm used here was very significantly different from the previously published one and not yet described elsewhere. We have now simplified the MAPIR description, only maintaining a general description and the parts which are useful for the discussion.

- *Section 2.2.4: this algorithm description can also be shortened, as Table 1 provides a reference and some details for the interested reader, and much of the information provided is not necessary for the interpretation of the results. For example P10L2-7, P10L18-22.*

The section has been shortened as suggested by the referee.

- *P13L10-24: from this it seems like the movement did not make a large systematic change to the dust height (0.02 km or less). Since the standard deviation of the height change was 0.25 km, this also implies that the error which would result from not attempting to account for the time difference is small, and that the altitude of dust layers is fairly stable over the time period of (as I understand) up to 5 hours. So that in itself is an interesting result as it suggests temporal sampling errors are not a big problem for this type of comparison, at least when looking at the big picture (since 0.25 km is somewhat smaller than the other retrieval uncertainties). The other side of the coin, which I think should be discussed more, is whether FLEX-TRAs assessment of the transport during this time period is accurate. I would like to see some quantification on the reliability of FLEXTRA here as I assume it depends in part on the resolution and quality of the meteorology data ingested, since this is what will determine the horizontal and vertical motion from the trajectory. So for example which ECMWF data version was used? Has FLEXTRA been quantitatively evaluated? If you use another meteorological data set or change some other input parameter is the solution stable? I know that e.g. HYSPLIT lets you run an ensemble of trajectories as one way to assess stability; I dont know if FLEXTRA does this too. The bottom line is that the comparability between CALIPSO and other sensors rests on the use of FLEXTRA to assess and correct for the change in aerosol location between observations, and this is not covered in much detail in the present version of the manuscript, so more discussion is needed.*

To address the referee's comments we have added the following to the FLEXTRA description: "Quantification of trajectory errors is always difficult, due to a general lack of ground-truth data. However, FLEXTRA has been quantitatively evaluated in the past. Comparisons of FLEXTRA trajectories driven with ECMWF data with balloon trajectories have revealed typical horizontal transport errors of about 20% of the travel distance, but with large variability from case to case (Baumann and Stohl, 1997; Stohl and Koffi, 1998; Riddle et al., 2006). Evaluation against meteorological tracers such as potential vorticity suggests errors of a similar magnitude (Stohl and Seibert, 1998). Thanks to improvements in the meteorological analysis data, slightly smaller errors may be assumed for more recent years, but the order of magnitude of the errors is likely still similar.".

Information about ECMWF data version and temporal and spatial resolution has been added.

- *Figures 2-7: I understand the intention here, but find these figures hard to interpret. In the top panel the grey is hard to see unless you zoom a lot, and the blue lines are somewhat similar to tones in the colour table. For the bottom panels, the background lidar curtain is the same between the IASI figures but there are too many different coloured symbols overlaid to quickly and easily compare and get the picture of what is going on. I wonder if, for the IASI plots at least, these could be redrawn. For example, make the top panels of Figures 2-5 their own figure, since these are all the same date and same map bounds, so we can directly compare the maps of coverage and heights between the different algorithms for IASI. Then also make the bottom panels their own figure and simplify in some way for clarity. For example we probably dont need to see the lidar curtain (although this could be added as a panel by itself at the top, or as a separate figure) since what is being compared is the effective heights, and the curtain just adds noise when the eye is trying to see coloured symbols. The deep pink pluses and red crosses appear similar in tone, again taking the lidar curtain off would allow one to use a more contrasting color for one such as blue or black. The points for CALIPSO AOD could also be removed since this is in black and naturally the most eye-catching, yet it is the least relevant since it is one set of points which does not represent a height. (In my mind the think you want the reader to focus on should probably be black or red since these stick out.) Alternatively the CALIPSO AOD could be just overlaid on the lidar curtain. The point here is that when looking at figures like this I want to answer two questions: (1) how similar are the spatial coverage and distribution of heights from the data sets and (2) how do they compare to CALIPSO? The present Figures do not allow me to do this effectively.*

To make the figures more readable, we have split the figures as suggested and changed the colours of the markers and included legends. The text as been updated accordingly.

- *I realise that Figures 6 and 7 are a different date so cant be combined with the IASI figures for my suggested redrawing. Is there no date with all instruments providing data? If so that would be better to show. Since both have much sparser coverage than the IASI plots, though (both these Figures are essentially a lot of white space*

*and then overlapping symbols which are hard to distinguish without zooming), unless a better common case can be found, perhaps these two examples could be removed. I understand that from parity you probably want to show one case from each algorithm, but surely there must be a more instructive case than this; there is so little data that its hard to assess looking at it whether these data sets are reasonable or not. And the summary statistics are in Table 3, so if the data are always sparse I dont know that we need to see GOME/SCIAMACHY case studies.*

We have omitted these figures as suggested. The text have been updated accordingly.

- *Table 3: as with Table 1, I'd add algorithm name here in the column headers. I'd also add instrument name, for ease of reference. I also have some formatting questions/comments here. In general this table is not well organised because the values in the headers only seem to refer to some of the statistics (I guess the top four of the eight boxes in each subset?) and there are too many colours. I think this could be much simplified, and the clarity improved, by condensing things and labelling individual rows rather than relying on colour-coding and large column headers. If I interpret the legend right, the top left and top right numbers in each column are the mean and standard deviation of height difference. So this could be represented as one entry, e.g. 0.590 +/- 1.213 km for BIRA-IASB. And the row legend would just read "height difference, km". Then youd have other rows for number of points and "inlay". Labelling rows (which would then repeat for the all points, day/night, land/ocean splits) would remove the need for the complicated colour coding, and the header would only say instrument and institute/algorithm name. The statistics for cumulative vs. geometric heights could be split as left-right subcolumns for each algorithm instead. Does this make sense? Changing to that layout would make each row/columns content unambiguous, and dis- pense with the need to have 10 different colours (plus white) to code the table. As it stands, again, it is very difficult to pick out the key numbers from the table.*

We agree with the referee that the table was unnecessarily complicated to read and appreciate the suggestions for changes. They have been fully adopted in the revised manuscript.

- *Figure 8 (and later): if these are scatter density plots on the first two rows then shouldnt they be shown as filled rectangles rather than clouds of points? There is also no colour scale to indicate what is shown (absolute or relative frequency, how many points are we looking at)? I suggest redrawing. For the bottom panels, it would be clearer if a vertical line for a height difference of 0 km is added. This will again aid in direct comparisons of the data sets. I would take off the Normal distribution fits; they dont add anything that we cant already see from the bins, and dont appear to be great fits in some cases anyway (the actual distributions seem to have higher kurtosis than a Normal distribution, at least for IASI).*

The two first rows in figure 8 (and later) show the probability density, using kernel density estimation. Two-dimensional histogram, which gives filled rectangles, does not reveal the features seen with the probability density. The caption of the figure has been changed to clarify what is presented. Also color scales have been added

in the two first rows, and in the bottom row the normal distribution fit has been removed and a vertical line added for the zero height difference, as suggested.

- *Figures 11, 12: I dont think these add anything to the discussion, and should be deleted. This is essentially another visualisation of the data already presented in Fig- ures 8-10 and Table 3, albeit also sliced by time. The lines are also too faint to see. The authors can just modify the discussion on P24, L12-16 to note that optical charac- teristics may have been different but no clear temporal variation was found when the data were examined. We dont need to see the plots.*

The figures have been removed and the text modified accordingly as suggested.

- *P24L19-23: From the inlay columns it seems overall that this has values between 5.3% (LISA, ocean, night, cumulative) and 68.9% (LMD, land, night, cumulative), with typical values being something of order 30%. That means that typically two thirds or so of retrieved heights are entirely outside the dust layer (i.e. the retrievals put the dust somewhere totally without dust). Is that interpretation really correct? If so, that sounds pretty bad, and seems at odds with the other statistics presented, which show a standard deviation of around 1 km or so (and to me seems like a good result). I would double-check the calculation of this inlay statistic or reword the text if I have misunderstood what it means. The only way I can think of to reconcile this discrepancy is if the vertical extent of the CALIPSO dust layers is typically significantly smaller than the roughly 1 km IASI retrieval error. So perhaps some statistics about the CALIPSO dust layer geometric thickness should be presented here. Either way, there appears to be some unresolved issue in this statistic or its interpretation.*

Unfortunately there was an error in the calculation of the "inlay" statistics. This error, which affected only the "inlay" statistics, has been corrected and revised results entered Table 3. The text has been updated accordingly. A table showing CALIOP cloud height and layer thickness statistics have been added together with an accompanying discussion in the text.

- *P27L14-15: Table 1 indicates that several of the algorithms use the Optimal Estima- tion Method (OEM) or similar techniques, which should be able to provide pixel-level uncertainty estimates on retrieved aerosol height. It would be instructive to compare these estimates in a statistical sense to the level of agreement with CALIPSO, to see whether these uncertainties are reasonable. This does not appear to have been done; I suggest adding it in the revised manuscript, since this analysis provides a useful way to validate the uncertainty estimates. I realise that this cant be done for all the data sets, but since this is a big advantage of OEM, it makes sense to use it! For the OEM retrievals, it should be there already. If it is not, why not? It is definitely naturally within the scope of the existing study.*

Some of the algorithms use methods such as the OEM which in principle may be used to provide pixel-level uncertainties. However, these uncertainties include only those associated with the quantities included in the retrieval and are not fully representative because we are not able to compute the full uncertainty estimate.

To compute the full uncertainty estimate would require computing derivatives with respect to the parameters with significant impact on the retrieval. For the MAPIR algorithm for example, this would at least include derivatives of the temperature profile, surface emissivity, aerosols size distribution and refractive index. Within the current MAPIR framework this is not possible. For the other algorithms similar concerns are valid, and/or full error estimates have not been included at this stage of algorithm development. Uncertainty estimates of the retrievals are certainly of great interest, but also not trivial to implement. At the current algorithm development stage we are unfortunately not able to present this quantity.

**Response to interactive comments from Referee #3**

The referee is thanked for the careful reading of and constructive comments to the manuscript. The referee's comments are repeated below in italic font. The responses to the comments are shown in roman font.

**Major comments**

1. *I have the same feeling to the first reviewer that Figure 2-7 and their discussions are difficult to follow. I also suggest Figures 2-5 and 6-7 be merged, so that readers can easily compare the spatial distribution of heights between different algorithms. I would also recommend a legend be added to the curtain plot to indicate the meanings of each symbol.*

   Figures 2-7 have been changed as suggested by referee #1. Legends have been added to indicate the meanings of each symbol.

2. *Table 3 is particularly hard to follow. I recommend, instead of using table, use bar plots to compare those statistics to different algorithms.*

   We have clarified and cleaned up Table 3 as suggested by referee #1. Bar plots, as suggested, may be an alternative. However, we feel that the numbers themselves include more detailed information about the results and, as such, may have more value for possible future studies.

3. *Too much text is used to present Figures in the Appendix. Those figures should be briefly mentioned, so only the major findings from them be presented.*

   We have moved the mentioned text to the appendix and points the reader to the appendix for discussion and presentation of these results.

4. *This study found that solar algorithms yield larger bias (> 1 km) for the case of dust aerosol height than the IR algorithms. However, it should be noted that some studies have shown an accuracy of about 0.5 km of dust layer height from O2-A might (studies listed below). So authors may need to compare and justify the performance of the current study to those studies:*

   *Kokhanovsky, A. A., and V. V. Rozanov (2010), The determination of dust cloud altitudes from a satellite using hyperspectral measurements in the gaseous absorption band, International Journal of Remote Sensing, 31(10), 2729-2744, doi:10.1080/01431160903085644.*

   *Dubuisson, P., R. Frouin, D. Dessailly, L. Dufor TM t, J.-F. . L on, K. Voss, and D. Antoine (2009), Estimating the altitude of aerosol plumes over the ocean from reflectance ratio measurements in the O2 A-band, Remote Sensing of Environment, 113(9), 1899-1911, doi:10.1016/j.rse.2009.04.018.*

*Xu, X., J. Wang, Y. Wang, J. Zeng, O. Torres, Y. Yang, A. Marshak, J. Reid, and S. Miller (2017), Passive remote sensing of altitude and optical depth of dust plumes using the oxygen A and B bands: First results from EPIC/DSCOVR at Lagrange-1 C2point, Geophysical Research Letters, 44(14), 7544-7554, doi:10.1002/2017GL073939.*

We thank the referee for these references. In the Discussion section we have included these references and a discussion of the results, highlighting additional differences between the retrieval setups of this work and mentioned literature.

**Specific comments:**

- *P2, L11: I'd to bring an attention to a recent review article about passive remote sens- ing of aerosol height by Xu et al. 2018, which is worth to cite: Xiaoguang Xu, Jun Wang, Yi Wang and Alexander Kokhanovsky, Chapter 1 - Passive Remote Sens- ing of Aerosol Height, In Remote Sensing of Aerosols, Clouds, and Precipitation, Elsevier, 2018, Pages 1-22, ISBN 9780128104378, https://doi.org/10.1016/B978-0-12-810437- 8.00001-3*

  The paper has been cited in the introduction.

- *P3, L5-10: It mentioned here that these selected dust events are of Saharan origin, but the studied area are also frequently affected by dust emitted from Middle East, India, and Western China. Please be accurate.*

  We have rephrased the sentences on P3, L5-12, to also include dust emitted from Middle East, India, and Western China.

- *P14, Figure 2(bottom): Symbols are hard to follow. A legend may be added to indicate the meaning of each symbol.*

  This Figure and similar ones have been revised. Legends have been added.

- *P15, Table 2: I dont quite understand the bracketed numbers in the third and forth rows. Please clarify in the Table caption (or using table footnote, as the caption is already very long).*

  We have added footnotes explaining the bracketed numbers.

- *P21, Figure 8: A colorbar is needed for the density of the scatters (similarly in Figure 9-10). The definition of the density is also necessary in the figure caption.*

  Colorbars have been added to Figure 8 and similar Figures. The definition of the density have been added to the figure caption.

**Response to interactive comments from Referee #3**

The referee is thanked for the careful reading of and constructive comments to the manuscript. The referee's comments are repeated below in italic font. The responses to the comments are shown in roman font.

**Major comments**

1. *I have the same feeling to the first reviewer that Figure 2-7 and their discussions are difficult to follow. I also suggest Figures 2-5 and 6-7 be merged, so that readers can easily compare the spatial distribution of heights between different algorithms. I would also recommend a legend be added to the curtain plot to indicate the meanings of each symbol.*

   Figures 2-7 have been changed as suggested by referee #1. Legends have been added to indicate the meanings of each symbol.

2. *Table 3 is particularly hard to follow. I recommend, instead of using table, use bar plots to compare those statistics to different algorithms.*

   We have clarified and cleaned up Table 3 as suggested by referee #1. Bar plots, as suggested, may be an alternative. However, we feel that the numbers themselves include more detailed information about the results and, as such, may have more value for possible future studies.

3. *Too much text is used to present Figures in the Appendix. Those figures should be briefly mentioned, so only the major findings from them be presented.*

   We have moved the mentioned text to the appendix and points the reader to the appendix for discussion and presentation of these results.

4. *This study found that solar algorithms yield larger bias (> 1 km) for the case of dust aerosol height than the IR algorithms. However, it should be noted that some studies have shown an accuracy of about 0.5 km of dust layer height from O2-A might (studies listed below). So authors may need to compare and justify the performance of the current study to those studies:*

   *Kokhanovsky, A. A., and V. V. Rozanov (2010), The determination of dust cloud altitudes from a satellite using hyperspectral measurements in the gaseous absorption band, International Journal of Remote Sensing, 31(10), 2729-2744, doi:10.1080/01431160903085644.*

   *Dubuisson, P., R. Frouin, D. Dessailly, L. Dufor TM t, J.-F. . L on, K. Voss, and D. Antoine (2009), Estimating the altitude of aerosol plumes over the ocean from reflectance ratio measurements in the O2 A-band, Remote Sensing of Environment, 113(9), 1899-1911, doi:10.1016/j.rse.2009.04.018.*

*Xu, X., J. Wang, Y. Wang, J. Zeng, O. Torres, Y. Yang, A. Marshak, J. Reid, and S. Miller (2017), Passive remote sensing of altitude and optical depth of dust plumes using the oxygen A and B bands: First results from EPIC/DSCOVR at Lagrange-1 C2point, Geophysical Research Letters, 44(14), 7544-7554, doi:10.1002/2017GL073939.*

We thank the referee for these references. In the Discussion section we have included these references and a discussion of the results, highlighting additional differences between the retrieval setups of this work and mentioned literature.

**Specific comments:**

- *P2, L11: I'd to bring an attention to a recent review article about passive remote sens- ing of aerosol height by Xu et al. 2018, which is worth to cite: Xiaoguang Xu, Jun Wang, Yi Wang and Alexander Kokhanovsky, Chapter 1 - Passive Remote Sensing of Aerosol Height, In Remote Sensing of Aerosols, Clouds, and Precipitation, Elsevier, 2018, Pages 1-22, ISBN 9780128104378, https://doi.org/10.1016/B978-0-12-810437- 8.00001-3*

  The paper has been cited in the introduction.

- *P3, L5-10: It mentioned here that these selected dust events are of Saharan origin, but the studied area are also frequently affected by dust emitted from Middle East, India, and Western China. Please be accurate.*

  We have rephrased the sentences on P3, L5-12, to also include dust emitted from Middle East, India, and Western China.

- *P14, Figure 2(bottom): Symbols are hard to follow. A legend may be added to indicate the meaning of each symbol.*

  This Figure and similar ones have been revised. Legends have been added.

- *P15, Table 2: I dont quite understand the bracketed numbers in the third and forth rows. Please clarify in the Table caption (or using table footnote, as the caption is already very long).*

  We have added footnotes explaining the bracketed numbers.

- *P21, Figure 8: A colorbar is needed for the density of the scatters (similarly in Figure 9-10). The definition of the density is also necessary in the figure caption.*

  Colorbars have been added to Figure 8 and similar Figures. The definition of the density have been added to the figure caption.

**Comparison of dust layer heights from active and passive satellite sensors**

[revised manuscript text omitted]

The remainder of the paper is organized as follows: In Sect. 2 the data and data analysis methods are presented. The results from the aerosol layer height comparison are given in Sect. 3. The results are discussed in Sect. 4 and followed by the conclusions.

**2 Data and methodology**

To allow inclusion of data from the SCIAMACHY instrument that ceased operation in 2012, four desert dust events  in 2010 were selected (total 40 days):

- 18-27 March (10 days)

- 22 May - 1 June (11 days)

- 1-12 July (12 days)

- 14-20 September (7 days)

 The comparison focus on the region between 0-40°N and 80°W-120°E, see Fig. 2, and is mainly affected by dust from Sahara, but is also influenced by dust from the Middle East, India and Western China.

**2.1 Active instrument dust height retrievals - CALIOP**

[revised manuscript text omitted]

data by the BIRA-IASB (top plot, Fig. **??**2), DLR (second plot, Fig. **??**2), LMD (third plot, Fig. **??**2), and LISA (bottom plot, Fig. **??**2) algorithms,  are overlaid by

[Figure]

**Figure 3.**  Curtain plot of the CALIOP extinction coefficent for heights identified as dust. The black dots are the column optical depth at 532 **??**nm from CALIOP. The curtain  is for the CALIOP data between 40-60°E in the top plot.

CALIOP cumulative extinction heights, derived from profiles identified as dust (step 3,  red dots), that are within the temporal and spatial requirements. CALIOP data are recorded after the IASI overpass. To account for possible movements of the dust between the overpasses the CALIOP dust heights were moved in longitude, latitude and height using the FLEXTRA model

5  (step 4, Stohl et al., 1995) . FLEXTRA does not include turbulence or loss processes.  FLEXTRA calculated mean-wind trajectories with meterological input data from the ECMWF. Here operational data with a 1° latitude × 1° longitude resolution, 91 vertical levels and a time resolution of 3 hrs were used. Quantification of trajectory errors is always difficult, due to a general lack of ground-truth data. However, FLEXTRA has been quantitatively evaluated in the past. Comparisons of FLEXTRA trajectories driven with ECMWF data with balloon

10 trajectories have revealed typical horizontal transport errors of about 20% of the travel distance, but with large variability from case to case (Baumann and Stohl, 1997; Stohl and Koffi, 1998; Riddle et al., 2006) . Evaluation against meteorological tracers such as potential vorticity suggests errors of a similar magnitude (Stohl and Seibert, 1998) . Thanks to improvements in the meteorological analysis data, slightly smaller errors may be assumed for more recent years, but the order of magnitude of the errors is likely still similar.

15 The black dots in Fig. 2 are CALIOP dust height pixels that have been moved from their original location ( red dots) to the nearest IASI pixel (step 5). As the cumulative extinction and geometric mean CALIOP dust heights are different they will be moved by FLEXTRA to different locations.  An example of this is seen in  Fig. 4 where the cumulative extinction (black circles) and geometric mean (red circles) heights from the passive instruments sometimes overlap (not moved or moved to same location and height) and sometimes do not overlap (moved

to different location and/or height). It is also seen in the difference in number of colocated  point, Table 2. For the full data period the CALIOP dust heights were on average moved upwards by 0.015 (cumulative extinction) and 0.020 km (geometric mean), both with standard deviations of 0.25 km.

**3  Results**

5  The analysis steps 1-5, section 2.3, were performed for all days and algorithms. The number of dust pixels identified by the various algorithms after step 1 is given in Table 2. The number of pixels identified as dust by the various IASI-algorithms varies by a factor of 4.6. The differences reflect the differences in dust detection methods and it is outside the scope of this study to further investigate the reasons for these differences. As expected the solar algorithms detect far fewer dust pixels due to only day time coverage (factor of 2) and larger pixels size (factor of about 16). The difference between the two solar algorithms (KNMI and IUP) are due to differences in the constraints set to detect dust. In step 2 dust pixels within a given

**Table 2.** The number of data points (dust heights) at data reducing step of the data analysis chain described in section 2.3.  Step number refers to the analysis steps as described in section 2.3.

|  Institute (Instrument/Algorithm) Step | BIRA-IASB (IASI/MAPIR) | DLR (IASI/IMARS) | LMD IASI | LISA (IASI/AEROIASI) | KNMI GOME-2 | IUP SCIAMACHY |
|---|---|---|---|---|---|---|
| 1 | 2324277 | 503944 | 811360 | 1770793 | 21535 | 2710 |
| 2[a] | 13377 (0.58) | 5208 (1.0) | 14916 (1.8) | 13110 (0.74) | 3715 (17.3) | 1979 (73.0) |
| 5-cumulative extinction[b] | 2620 (19.6) | 1420 (27.3) | 748 (5.0) | 2203 (16.8) | 215 (5.8) | 34 (1.7) |
| 5-geometric mean[b] | 2408 (18.0) | 1296 (24.9) | 704 (4.7) | 1978 (15.1) | 91 (2.4) | 21 (1.1) |

a: Numbers in parenthesis are data points in percentage relative to the total number in the previous analysis step.

b: Numbers in parenthesis are data points in percentage relative to the total number in analysis step number 2, for example 18.0=2408/13377 (column 2, row 4).

[revised manuscript text omitted]

**Table 3.** The mean  ± the standard deviation of the dust height difference between the passive sensors and CALIOP.  , and the number (#) of colocated points. The inlay  is the percentage of heights that are within the CALIOP layer.  For BIRA-IASB, DLR, LISA and LMD statistics are given for all  data and subgroups of data recorded during day and night and over land  and ocean. For KNMI-GOME2 only day comparisons are possible, hence the lack of comparisons with CALIOP night overpasses.  Aslo note that for KNMI-GOME2 the number of land and ocean pixels does not add up to the total due to some pixels covering coastal regions (mixed pixels).

| Institute | BIRA-IASB | DLR | LMD | LISA | |
|---|---|---|---|---|---|
| Instrument/Algorithm |  IASI/MAPIR |  IASI/IMARS |  IASI |  IASI/AEROIASI |  |
|  height | *CALIOP Day and Night, cumulative extinction heights* | | | | |
| Height difference (km) | 0.590±1.213 | 0.785±1.281 | -0.053±1.339 | 0.507±1.126 | |
|  points (#) | 2620 | 1420 | 748 | 2203 | |
|  inlay (%) | 83.1 | 78.3 | 77.5 | 85.8 | |
| | *CALIOP Day and Night, geometric mean heights* | | | | |
| Height difference (km) | 0.078±1.108 | 0.243±1.181 | -0.607±1.187 | -0.045±1.029 | |
| points (#) | 2408 | 1296 | 704 | 1978 | |
| inlay (%) | 81.1 | 77.5 | 75.9 | 84.0 | |
| | *CALIOP Day, Land, cumulative extinction heights* | | | | |
|  Height difference (km) | 0.357±1.665 | 0.405±1.660 | -0.102±1.448 | -0.225±1.454 | |
| points (#) | 605 | 377 | 319 | 440 | |
| inlay (%) | 58.5 | 61.0 | 70.2 | 71.4 | |
| | *CALIOP Day, Land, geometric mean heights* | | | | |
| Height difference (km) | 0.087±1.572 | -0.044±1.526 | -0.496±1.322 | -0.635±1.357 | |
| points (#) | 598 | 393 | 322 | 425 | |
| inlay (%) | 57.7 | 59.8 | 70.5 | 70.1 | |
| | *CALIOP Day, Ocean, cumulative extinction heights* | | | | |
| Height difference (km) | 0.783±0.913 | 0.913±1.539 | -0.501±1.409 | 0.172±1.389 | |
|  points (#) | 172 | 22 | 118 | 180 | |
| inlay (%) | 74.4 | 59.1 | 58.5 | 62.2 | |
| | *CALIOP Day, Ocean, geometric mean heights* | | | | |
| Height difference (km) | 0.340±1.187 | 0.184±1.174 | -0.922±1.142 | -0.285±1.187 | |
| points (#) | 170 | 22 | 109 | 170 | |
| inlay (%) | 72.4 | 68.2 | 55.0 | 59.4 | |
| | *CALIOP Night, Land, cumulative extinction heights* | | | | |
| Height difference (km) | 0.567±1.020 | 0.906±1.062 | 0.073±1.092 | 0.663±0.896 | |
| points (#) | 1501 | 996 | 206 | 1226 | |
| inlay (%) | 91.0 | 84.8 | 92.2 | 91.2 | |
| | *CALIOP Night, Land, geometric mean heights* | | | | |
|  Height difference (km) | 0.038±0.903 | 0.358±0.964 | -0.579±1.058 | 0.170±0.855 | |
| points (#) | 1330 | 854 | 177 | 1064 | |
| inlay (%) | 89.4 | 85.4 | 89.8 | 89.6 | |
| | *CALIOP Night, Ocean, cumulative extinction heights* | | | | |
| Height difference (km) | 1.008±0.741 | 1.599±1.127 | 0.352±1.180 | 1.043±0.637 | |
| points (#) | 342 | 25 | 105 | 357 | |
| inlay (%) | 96.5 | 96.0 | 92.4 | 97.2 | |
| | *CALIOP Night, Ocean, geometric mean heights* | | | | |
| Height difference (km) | 0.094±0.678 | 0.835±0.720 | -0.674±0.878 | 0.152±0.486 | |
| points (#) | 310 | 27 | 95 | 319 | |
|  inlay (%) | 95.8 | 96.3 | 91.7 | 96.9 | |

**Table 4.**

CALIOP mean cumulative and geometric mean dust layer height± standard deviation together with the dust layer thickness± standard deviation. Statistics are given for the full dataset and for subsets divided in the land and ocean for day and night overpasses.

[revised manuscript text omitted]

15 aerosol height from POLarization and Directionality of the Earth's Reflectances (POLDER) and MEdium Resolution Imaging Spectrometer (MERIS) oxygen A-band measurements. They showed that aerosol height estimates vary with aerosol optical depth (AOD), single scattering albedo, aerosol phase function, aerosol layer height, and the underlying surface albedo. For low surface albedo theoretical analysis gave errors of about $\pm 0.5$ km and $\pm 0.2$ km for POLDER and MERIS, respectively. Comparison between POLDER and CALIOP gave standard deviations less than about 0.55 km consistent with the theoretical
20 analysis. However, for parts of the three cases of coast Africa, the aerosol height was underestimated by up to 1-2 km . Dubuisson et al. (2009) attributed this to either a more complex vertical aerosol structure including a layer near the surface or the presence of low clouds under the aerosol layer. The theoretical sensitivity results of Dubuisson et al. (2009) was confirmed by Kokhanovsky and Rozanov (2010) whose modelling sensitivity study indicated aerosol heights within $\pm 0.5$ km if the aerosol single scattering albedo used in the retrieval deviated by less then 0.01 from the actual value of 0.99. They also
25 reported that error increased with dust layer top height.

It must be stressed that the reported errors on dust height, based on synthetic sensitivity studies, are estimated with the assumption that either AOD, optical model, or surface reflectance are perfectly known beforehand or are derived from independent instrument channels. For instance, Dubuisson et al. (2009) make first use of the official POLDER and MERIS AOD products, while focusing on dark ocean surfaces only. Then, they find the most accurate aerosol model (being this pure dust or a mixture
30 with sea salt or biomass burning) by perturbing the reflectance in a non-absorbing channel. Conversely, the solar algorithms of this work have been designed to fit the oxygen spectrum to concurrently infer dust height and optical thickness together, so that AOD and height uncertainties cannot be decoupled and deviations of the assumed optical model and climatological surface reflectivity from the actual ones contribute to the overall error budget. As such, the evaluation of the presented dust cases can be regarded as a more comprehensive test bed for operational dust height retrievals.

A similar approach has been devised by Xu et al. (2017) . Combining oxygen A and B-band measurements from the Earth Polychromatic Imaging Camera (EPIC) on the Deep Space Climate Observatory (DSCOVR), they retrieved AOD and altitude, finding that 71.5% and 98.7% aerosol heights were respectively within $\pm 0.5$ km and $\pm 1.0$ km envelopes, when compared to CALIOP for two overpasses of Saharan dust events over water only. Their reported RMS error, in this case, amounts to 0.45 km, pointing to the advantage of adding in the retrieval the information concealed in the B-band.

Generally, we found the IUP-SCIAMACHY and KNMI-GOME-2 algorithm retrieved heights to be low by -1.097 km (-0.961) and -1.393 km (-0.818) respectively when compared with CALIOP geometric mean (cumulative extinction) heights. For the KNMI-GOME-2 algorithm the underestimate is larger over ocean -2.015 km (-1.477) than over land -0.893 km (-0.229), Table 3. These differences are larger than those reported in the above cited studies. Still, it is found that for KNMI-GOME-2 (IUP-SCIAMACHY) between 63.7-67-0% (40.9-45.7%) of the aerosol heights are within the CALIOP aerosol layer. For KNMI more of the retrieved heights are within the CALIOP aerosol layer over land than ocean, Table 3.

[revised manuscript text omitted]

Zhou, D., Larar, A., Liu, X., Smith, W., Strow, L., Yang, P., Schlüandssel, P., and Calbet, X.: Global Land Surface Emissivity Retrieved From Satellite Ultraspectral IR Measurements, Geoscience and Remote Sensing, IEEE Transactions on, 49, 1277–1290,

https://doi.org/10.1109/TGRS.2010.2051036, http://ieeexplore.ieee.org/stamp/stamp.jsp?tp=&arnumber=5523979&isnumber=5738422, 2011.

[Figure]

**Figure 4.**  IASI dust  layer heights colocated to CALIOP cumulative extinction heights (black circles) and CALIOP geometric mean heights (red circles) for the same time  and location as in Fig. 3.  Also shown are shifted (upward triangles) and unshifted (downward triangles) CALIOP cumulative extinction and geometric mean heights.

[Figure]

**Figure 5.** (Upper row)  The probability density, using kernel density estimation, of the CALIOP cumulative extinction height versus height from the various algorithms.  Also given are the Pearson's correlation coefficient and root mean square error (RMSE). (Centre row) The probability density, using kernel density estimation, of the difference between the passive algorithm and the CALIOP cumulative extinction heights versus the CALIOP column extinction.  (Bottom row) Frequency distribution of the difference between the height from the various algorithms and the CALIOP cumulative extinction height.  The mean and standard deviation ($\sigma$)  together with the number data points are given in each plot. This information is also provided in Table 3. Data are shown for the BIRA-IASB (first column), DLR (second column), LISA (third column) and LMD (fourth column) algorithms.

[Figure]

**Figure 6.** Similar to Fig. 5, but for the CALIOP geometric mean height .

[Figure]

**Figure 7.** Similar to Figs. 5-6, but for the KNMI (first and third column) and IUP (second and fourth column) algorithms versus the CALIOP cumulative extinction (first and second column) and geometric mean (third and fourth column) dust heights.

**Appendix A:  Additional figures**

In Figs. A1-A4 statistics are shown for all data, and land-day, ocean-day, land-night and land-day subsets for all IASI dust height retrieval methods as compared with the CALIOP geometric mean heights. Figs. A5-A8 show similar data but using the CALIOP cumulative extinction heights.

The plots in Figs. A1-A8 reflect the findings presented in Table 3. The BIRA-IASB algorithm agrees well with the CALIOP geometric mean height over land for day and night, Fig. A1. Over ocean the agreement is better during the night. It is noted that for the ocean day subset the histogram is bimodal. When compared with the cumulative extinction height, Fig. A5, the BIRA-IASB dust height is overestimated over ocean during both day and night; the ocean day subset appears to be bimodal; the agreement appears better over land during day than night, but this may in part be due to a bimodal histogram for the land day subset. This is reflected in the spread in the difference which is smaller during night than day. For DLR, Figs. A2 and A6, there are few data points available over the ocean. Over land the DLR height data are clumped at a single height for the day subset and at two heights for the night data subset. For LMD, Figs. A3 and A7, the agreement is mono-modal for the land day, land night and ocean night subsets when compared with both the CALIOP cumulative extinction and geometric mean heights. For the ocean day subset a bimodal distribution may be present. Overall the agreement is better when compared with the cumulative extinction height. The LISA data, Figs. A4 and A8, also have a bimodal ocean day distribution compared with the CALIOP heights. For the ocean the mean difference with the CALIOP cumulative extinction height is significantly larger during night than day. This difference is nearly a factor 2 smaller when comparing with the geometric mean height. For land the magnitude of the difference is smallest when compared with the cumulative extinction height during the day and with the geometric mean height during night.

[Figure]

**Figure A1.** (Upper row) Scatter plots of the CALIOP geometric mean height versus height from the BIRA-IASB algorithm. (Centre row) Scatter plot of the difference between the BIRA-IASB and CALIOP heights versus the CALIOP column extinction. (Bottom row) Frequency distribution of the difference between the height from the BIRA-IASB algorithm and the CALIOP height.  The mean and standard deviation of the normal distribution are given in each plot. (Column 1) All data points. (Column 2) CALIOP day data and IASI land pixels. (Column 3) CALIOP day data and IASI ocean pixels. (Column 4) CALIOP night data and IASI land pixels. (Column 5) CALIOP night data and IASI ocean pixels.

[Figure]

**Figure A2.** Similar to Fig. A5 but for the DLR algorithm.

[Figure]

**Figure A3.** Similar to Fig. A5 but for the LMD algorithm.

[Figure]

**Figure A4.** Similar to Fig. A5 but for the LISA algorithm.

[Figure]

**Figure A5.** (Upper row) Scatter plots of the CALIOP cumulative extinction height versus height from the BIRA-IASB algorithm. (Centre row) Scatter plot of the difference between the BIRA-IASB and CALIOP heights versus the CALIOP column extinction. (Bottom row) Frequency distribution of the difference between the height from the BIRA-IASB algorithm and the CALIOP height.  The mean and standard deviation of the normal distribution are given in each plot. (Column 1) All data points. (Column 2) CALIOP day data and IASI land pixels. (Column 3) CALIOP day data and IASI ocean pixels. (Column 4) CALIOP night data and IASI land pixels. (Column 5) CALIOP night data and IASI ocean pixels.

[Figure]

**Figure A6.** Similar to Fig. A5 but for the DLR algorithm.

[Figure]

**Figure A7.** Similar to Fig. A5 but for the LMD algorithm.

[Figure]

**Figure A8.** Similar to Fig. A5 but for the LISA algorithm.